# X-ray flares from the stellar tidal disruption by a candidate supermassive black hole binary

Xinwen Shu [1✉], Wenjie Zhang[1], Shuo Li[2], Ning Jiang [3], Liming Dou [4], Zhen Yan[5], Fu-Guo Xie[5], Rongfeng Shen [6], Luming Sun [1], Fukun Liu[7,8] & Tinggui Wang[3]

Optical transient surveys have led to the discovery of dozens of stellar tidal disruption events (TDEs) by massive black hole in the centers of galaxies. Despite extensive searches, X-ray follow-up observations have produced no or only weak X-ray detections in most of them. Here we report the discovery of delayed X-ray brightening around 140 days after the optical outburst in the TDE OGLE16aaa, followed by several flux dips during the decay phase. These properties are unusual for standard TDEs and could be explained by the presence of supermassive black hole binary or patchy obscuration. In either scenario, the X-rays can be produced promptly after the disruption but are blocked in the early phase, possibly by a radiation-dominated ejecta which leads to the bulk of optical and ultraviolet emission. Our findings imply that the reprocessing is important in the TDE early evolution, and X-ray observations are promising in revealing supermassive black hole binaries.

---

[1] Department of Physics, Anhui Normal University, Wuhu, Anhui 241002, China. [2] National Astronomical Observatories, Chinese Academy of Sciences, Beijing 100101, China. [3] CAS Key Laboratory for Researches in Galaxies and Cosmology, Department of Astronomy, University of Science and Technology of China, Hefei, Anhui 230026, China. [4] Department of Astronomy, Guangzhou University, Guangzhou 510006, China. [5] Shanghai Astronomical Observatory, Chinese Academy of Sciences, Shanghai 200030, China. [6] School of Physics and Astronomy, Sun Yat-Sen University, Zhuhai 519082, China. [7] Department of Astronomy, Peking University, Beijing 100871, China. [8] Kavli Institute for Astronomy and Astrophysics, Peking University, Beijing 100871, China. ✉email: xwshu@ahnu.edu.cn

Almost all massive galaxies appear to contain a central supermassive black hole (SMBH) with a mass of $\gtrsim 10^6 M_\odot$ (where $M_\odot$ refers to the solar mass)[1], yet most of them remain unobservable due to the lack of enough radiative output through accretion process. Stars that pass sufficiently close to a SMBH can be torn apart when the tidal force of the SMBH exceeds its self-gravity. Although roughly half of the stellar material will be ejected, the other half will remain bound and eventually be accreted, producing a luminous flare of electromagnetic radiation[2]. These tidal disruption events (TDEs) not only provide novel means of probing SMBH in otherwise quiescent galaxies[3] but also serve as a unique laboratory for studying the formation and evolution of accretion disk (e.g., see refs. [4–6]), the launch of relativistic jets[7,8], as well as gravitational wave (GW) emission through coalescence of SMBH binaries[9,10].

Early theoretical works predict that TDEs should produce a bright thermal emission peaking mainly in soft X-ray bands, which originates from a newly formed accretion disk[2,11]. The effective temperature of thermal radiation produced by disk accretion[12] is $T_{eff} \approx 4 \times 10^5 M_6^{-1/4}$ K, where $M_6 = M_{BH}/(10^6)\,M_\odot$. The TDEs discovered in the ultraviolet (UV) and optical bands, however, are found to have surprisingly low blackbody temperature (of $\sim 1 - 4 \times 10^4$ K) and correspondingly large blackbody radii ($\sim 10^{14-15}$ cm)[13], which are difficult to reconcile with the predicted radiation from a compact accretion disk[3,11]. The UV/optical emission can be instead powered by shocks from stream self-collisions during the formation of disk[4,14], or conversely thermal reprocessing of accretion power by a layer of gas at large radii[15,16]. X-ray observations within the first few months of discovery are critical to disentangle the dominant emission mechanisms of optical light, yet only a handful of optical TDEs have been successfully detected with X-ray emission[5,6,17,18]. The origin of dominant UV/optical emission and its association with the X-ray one in TDEs still remains unclear.

Here we show delayed X-ray brightening by a factor of >60, ~140 days after the optical flare, in the TDE OGLE16aaa, followed by several dips of X-ray emission during the afterwards decay phase. These properties are unusual among standard TDEs and are instead consistent with either the tidal disruption by SMBH binary (SMBHB) or changes in absorption along the line of sight. In this context, the X-ray non-detections at the very early times could be due to obscuration, in which the reprocessed accretion radiation may power the bulk of observed UV and optical emission.

## Results

**Optical and UV light curve analysis.** The optical transient, OGLE16aaa ($RA_{J2000} = 01h07m20.88s$, $DEC_{J2000} = -64deg16'20.7"$), was discovered by the Optical Gravitational Lensing Experiment (OGLE-IV[19]) and its Transient Detection System[20] on 2 January 2016 at a redshift of $z = 0.1655$, coincident with the nucleus of its host galaxy[21]. The I-band light curve shows a rise over ~30 rest-frame days of about 3 mag above the quiescent brightness of the host galaxy, reaching a peak I-band magnitude of 18.98 mag on 20 January 2016, and subsequently declining by ~0.6 mag (to $I = 19.57$ mag) over 2 weeks. Then, its flux starts to increase again at $t \approx 20$ days with respect to the initial peak, reaching a secondary maximum of $I = 19.33$ mag within 1 week, with evidence for another decline until $t = 50$ days (Fig. 1a). After a time interruption in the optical observations, the light curve appears to show a plateau since $t = 146$ days, with a flux comparable to that of pre-flare phase, suggestive of the dominance of the host emission. Following the initial optical peak, OGLE16aaa was monitored at three UV bands (UVW2, $\lambda_{eff} = 1928$ Å; UVM2, $\lambda_{eff} = 2246$ Å; UVW1, $\lambda_{eff} = 2600$ Å) by the Ultraviolet and

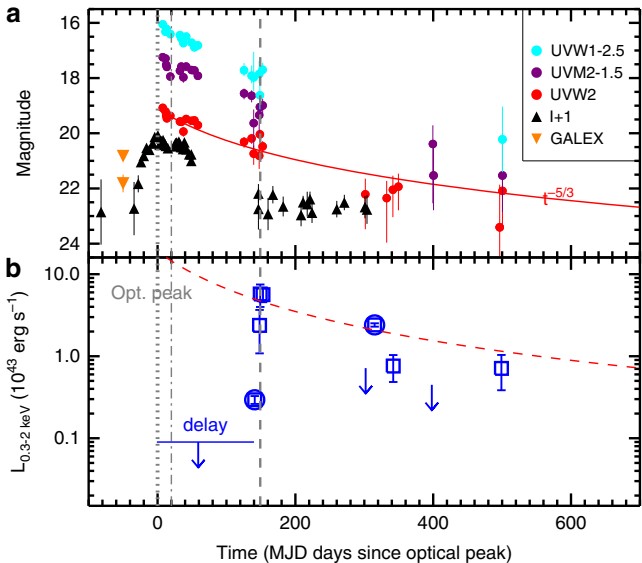

**Fig. 1 UV-optical and X-ray light curve of OGLE16aaa. a** The host-subtracted, Galactic dust extinction-corrected UV and optical light curve. The Swift UVW1, UVM2, and UVW2 photometric data are shown in cyan, dark purple, and red, respectively. We also plot the pre-flare GALEX UV data for comparison (orange). The optical I-band data are shown in black triangles. The light curves are shifted by constants for clarity as noted in the legend. All magnitudes are in the AB system. Error bars represent $1\sigma$ uncertainties due to photometric errors. The red line represents the fit to the UVW2 data with a canonical $n = -5/3$ power-law decay, assuming the same peak time as optical one[21] that is marked with gray dotted line. The dot-dashed line represents the approximate time of optical rebrightening, $t \approx 20$ days with respect to the initial peak. The dashed line marks the time for the X-ray peak. It is noteworthy that the true peak time might be slightly later, as no further observations are performed between the current peak and the non-detection at $t = 302$ days. **b** The follow-up X-ray observations from Swift (square) and XMM-Newton (circle). The $3\sigma$ upper limits on the flux for non-detections (see Table 1 and text) are shown with downward pointing arrows. The X-ray emission shows a clear time delay with respect to optical by ~140 days and brightened to peak within only ~2 weeks. All the X-ray luminosities are corrected for the absorption. Error bars represent $1\sigma$ uncertainties calculated using Monte Carlo simulations provided in XSPEC. The red dashed line corresponds to the best-fit to UVW2 data in the upper panel, but scaled to the UV/optical blackbody luminosity at $t = 153$ days (Supplementary Fig. 4).

Optical Telescope (UVOT)[22], as well as the X-ray band by the X-ray Telescope (XRT)[23] on board the Swift observatory[24] (Supplementary Table 1). We have analyzed all publicly available Swift observations (see "Methods": X-ray and UV data reduction). By including more data after March 2016, results of our reanalysis are generally consistent with those reported in the literature[21] and confirm that the UV emission decayed as expected from a TDE. Although the luminosity evolution can be described by the canonical $t^{-5/3}$ power model, the exponential decline model ($L = L_0 e^{-(t-t_0)/\tau}$) is able to fit the data equally well (Supplementary Fig. 2). As the UV emission appears to decay to a plateau level, we cannot distinguish between the two models with current observations. It is noteworthy that there is a tentative evidence for the UVW2 and UVM2 bands displaying a similar rebrightening as the optical. However, it is not statistically significant (Supplementary Note 2). In comparison with the GALEX[25] data taken on 2003, the UV emission at ~2300 Å near-UV (NUV) increased by a factor of 7.7, reaching a recorded peak luminosity as high as $10^{44}$ ergs$^{-1}$.

**Table 1 X-ray observations of OGLE16aaa.**

| Telescope | obsID | obs. date | Days | Exposure s | Counts rate $10^{-2}$ cts s$^{-1}$ | Flux $10^{-13}$ erg cm$^{-2}$ s$^{-1}$ |
|---|---|---|---|---|---|---|
| Swift-XRT | 00034281001-16 | 19 Jan 2016–8 June 2016 | 0–141 | 30,160 | <0.023 | <0.12 |
| XMM-PN | 0790181801 | 9 June 2016 | 141 | 10,550 | 1.01 ± 0.11 | 0.40 ± 0.05 |
| Swift-XRT | 00034281018 | 16 June 2016 | 148 | 619 | 0.60 ± 0.32 | 3.15 ± 1.72 |
| Swift-XRT | 00034281019 | 17 June 2016 | 149 | 797 | 1.44 ± 0.44 | 7.59 ± 2.31 |
| Swift-XRT | 00034281020 | 21 June 2016 | 153 | 1974 | 1.42 ± 0.27 | 7.46 ± 1.42 |
| Swift-XRT | 00034281021 | 17 Nov 2016 | 302 | 1684 | <0.18 | <0.94 |
| XMM-PN | 0793183201 | 30 Nov 2016 | 316 | 21,830 | 9.37 ± 0.21 | 3.18 ± 0.15 |
| Swift-XRT | 00034281024-26 | 18 Dec 2016–4 Jan 2017 | 333–350 | 4842 | 0.19 ± 0.07 | 1.01 ± 0.36 |
| Swift-XRT | 00034281027-29 | 19 Feb 2017–23 Feb 2017 | 397–401 | 2665 | <0.11 | <0.59 |
| Swift-XRT | 00034281030-31 | 31 May 2017–4 June 2017 | 497–501 | 3014 | 0.18 ± 0.08 | 0.94 ± 0.42 |
| Swift-XRT | 00034281032 | 9 Feb 2020 | 1481 | 1903 | <0.16 | <0.84 |

For non-detections in either individual epochs or combined data, the corresponding $3\sigma$ upper limits on the counts rate and flux are given. The X-ray counts rate and flux is in the 0.3–2 keV, respectively. The X-ray flux is corrected for the internal and Galactic gas absorption.

**X-ray flux evolution**. Despite frequently monitored by the XRT, no X-ray emission was detected in either individual or stacking observations in early times from 19 January to 8 June 2016 (Fig. 1b), with an 0.3–2 keV luminosity $L_X < 8.9 \times 10^{41}$ ergs$^{-1}$ ($3\sigma$ upper limit, assuming a blackbody spectrum of temperature of $kT_{BB} = 60$ eV, see the spectral analysis below). Hereafter, we defined the non-detection as <1 net count, or the probability of having source counts from the background >0.05 in Poisson statistics, and accordingly reported three net counts as the $3\sigma$ upper limit on the flux of interest. The source was first detected by deeper XMM-Newton[26] observation on 9 June 2016 (XMM1), ~141 days later since the optical peak, with $L_X = 2.9 \times 10^{42}$ ergs$^{-1}$. About a week later, Swift caught the source at an even higher luminosity, reaching an X-ray peak at $L_X \sim 7 \times 10^{43}$ ergs$^{-1}$ between 17 June and 21 June 2016. The luminosity then decreased by more than an order of magnitude in the following XMM-Newton and Swift observations. Details of X-ray observations are shown in Table 1.

The overall long-term evolution of the X-ray luminosity is unique among known X-ray detected TDEs, in particular the fast rise to the peak within only 2 weeks. Another striking feature in the X-ray light curve is that the source became completely invisible to Swift on 17 November 2016 after the peak, with a $3\sigma$ upper limit on the flux of $9.4 \times 10^{-14}$ erg cm$^{-2}$ s$^{-1}$ in the 0.3–2 keV (or equally $L_X < 7.2 \times 10^{42}$ ergs$^{-1}$). Then, it recurred to a much higher flux level in the second XMM observation on 30 November 2016 (XMM2), followed by a new flux dip. The timescale of the flux increase (about 2 weeks) is somewhat consistent with that observed in the earlier brightening epoch at ~140 days. We argue that the Swift non-detection before XMM2 is significant. Assuming mild source variability in standard TDE evolution and so the same flux as that obtained with the XMM2 observation, we would expect around ten counts in the 0.3–2 keV to be detected by Swift for an exposure of 1.7 ks. For a Poisson distribution, the chance of detecting <1 photon in the Swift observation is $5 \times 10^{-5}$. Conversely, we detected ~5–9 net counts in two subsequent Swift observations. The probability of having these counts from the background is <0.15% in Poisson statistics (or at a significance level of 99.85%). It is noteworthy that previous study of this event[21,27] either failed to detect the X-ray emission at early times (around optical flare) or used only the partial observations from the XMM-Newton and Swift datasets to describe the X-ray evolution up to 17 November 2016 (~300 days since optical flare). Our results are broadly consistent with the recent study[28], confirming the rapid X-ray brightening in

OGLE16aaa. However, ref. [28] does not identify further flux flares and dips in the decay phase, which are crucial to lead to the advanced interpretation of overall observing properties of the source that will be presented in the next section.

**X-ray spectral analysis**. Another notable property for the source is the quasi-soft X-ray spectra that lack emission above ~2 keV, typical for TDEs detected in the X-ray bands. This is most evident in the two XMM-Newton observations that have the best spectral quality. The spectrum obtained from the first XMM-Newton observation can be well fitted with a single blackbody component (zbbody in XSPEC) with a temperature of $kT_{BB} = 58 \pm 5$ eV, modified by the Galactic absorption ($N_H = 2.7 \times 10^{20}$ cm$^{-2}$), as shown in Fig. 2a. No additional absorption intrinsic to the source was required. We used the same single blackbody model to describe the X-ray spectrum in the second XMM-Newton observation and found a similar temperature for the blackbody emission (Fig. 2c). The data to model ratios, however, show a clear excess of emission at energies above ~0.7 keV (Fig. 2b), suggesting that the single blackbody model is not enough to describe the X-ray spectrum of XMM2. The spectral fitting result is improved significantly by the addition of a power-law component to the above model (Fig. 2f). The overall $\chi^2$ decreased by 46 for two extra parameters, with a $F$-test probability of $3.6 \times 10^{-9}$. In this case, we obtain a best-fitted blackbody temperature of $kT_{BB} = 60^{+9}_{-11}$ eV, which is still comparable to that obtained from XMM1. Albeit with large uncertainties, the additional power-law component is steep with a photon index $\Gamma = 6.4 \pm 0.5$, that is rarely seen in TDEs[27]. Alternatively, the excess emission can be described by another blackbody component with a higher temperature of $kT_{BB} = 90^{+20}_{-13}$ eV (Fig. 2d, e). In this case, the temperature for the primary blackbody component decreases slightly to $kT_{BB} = 51^{+5}_{-8}$ eV. Results of X-ray spectral analysis are presented in Table 2. Similar spectral analysis was also performed for the Swift data. Unfortunately the spectral signal-to-noise ratios are not sufficient for a precise determination of the temperature even after stacking the data from individual observations. In the Supplementary Fig. 3, we present the combined X-ray spectrum obtained by Swift at the peak. The majority of the counts for the combined spectrum fall into the low-energy range of 0.3–0.7 keV, indicating the spectrum has remained soft. The spectrum can be described by a blackbody model with $kT_{BB} = 73^{+22}_{-32}$ eV, consistent with the results observed with XMM-Newton within errors.

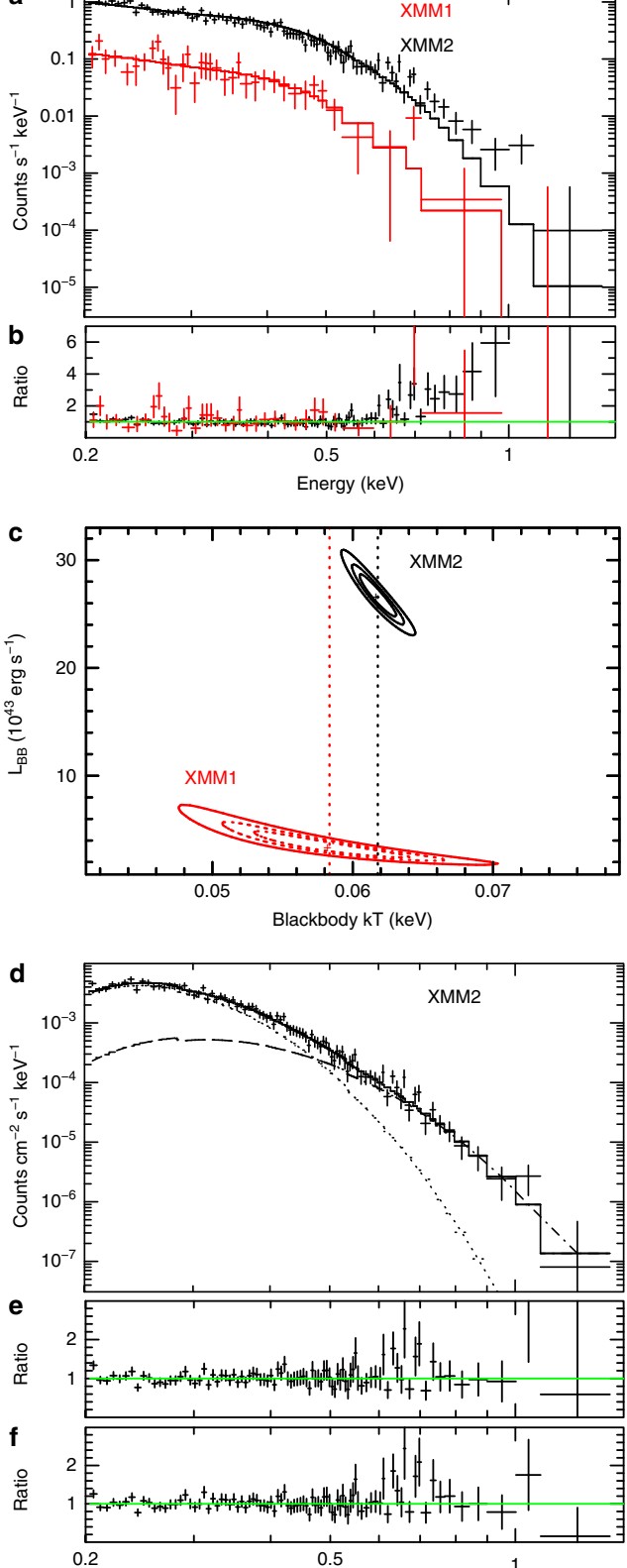

**Fig. 2 Fittings to the X-ray spectra of OGLE16aaa from XMM-Newton observations at two epochs. a** XMM-Newton PN spectra for OGLE16aaa along with the best-fitted single blackbody model. Error bars represent $1\sigma$ uncertainties calculated using Poisson statistics. The data observed on 9 June 2016 (XMM1) are shown in red, whereas that obtained on 30 November 2016 (XMM2) shown in black. The corresponding data to model radios are shown in **b**. **c** The joint 68%, 90%, and 99% confidence contours of the blackbody temperature vs. luminosity for the two observations. The vertical dotted lines represent the best-fitted temperatures. **d** The best-fitting result by including an additional blackbody component to account for the excess emission at above ~0.7 keV for the XMM2 data, which is shown in dashed line, and **e** is the corresponding data/model ratio. **f** The same as **e**, but for the data/model ratio from spectral fitting by using an additional power-law component to describe the excess emission.

**Table 2 Spectral fitting results for the X-ray data.**

| Model component | Parameter | XMM1 June 2016 | XMM2 Nov 2016 |
|---|---|---|---|
| Model 1 | | | |
| Blackbody | $kT_{BB}$ (eV) | $58 \pm 5$ | $60^{+9}_{-11}$ |
| Power-law | $\Gamma$ | ... | $6.4 \pm 0.5$ |
| Neutral absorber | $N_H$ ($\times 10^{22}$ cm$^{-2}$) | <0.06 | <0.23 |
| Model 2 | | | |
| Blackbody 1 | $kT_{BB1}$ (eV) | ... | $51^{+5}_{-8}$ |
| Blackbody 2 | $kT_{BB2}$ (eV) | ... | $90^{+20}_{-13}$ |
| Neutral absorber | $N_H$ ($\times 10^{22}$ cm$^{-2}$) | ... | <0.03 |
| Statistics | $\chi^2$/dof | 25.6/35 | 80.6/86 (82.2/87) |

The $\chi^2$ statistics in the parentheses is for Model 2.

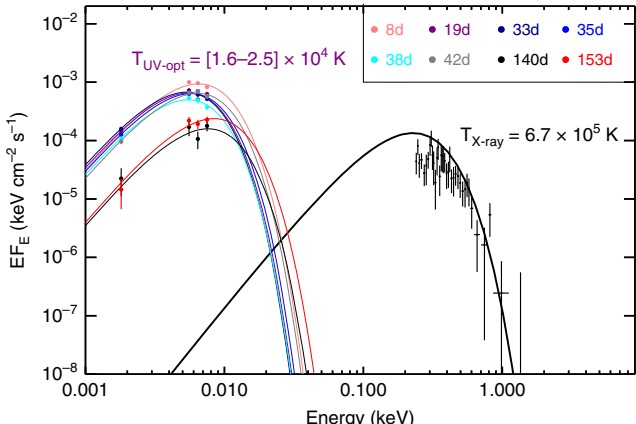

**Fig. 3 Evolution of the UV to optical SED of OGLE16aaa at different epochs.** The corresponding best-fitting blackbody models are shown in different colors (noted in the legend), with temperatures in the range [1.6 −2.5] $\times 10^4$ K. The epochs refer to the time relative to the initial optical peak, in units of days (d). The quasi-simultaneous X-ray spectrum observed with XMM-Newton (XMM1) about 140 days after the initial optical peak is shown in black, which can be fitted with a single blackbody model of $kT_{BB} = 58$ eV ($T_{X-ray} = 6.7 \times 10^5$ K) modified by Galactic absorption (see text). Error bars represent $1\sigma$ uncertainties that are same as that in Figs. 1 and 2. The unabsorbed blackbody model corrected for Galactic absorption is shown with black curve. Although with a much higher temperature, the best-fit blackbody to the X-ray data severely underpredicts the UV and optical flux, suggestive of different physical origins.

**Optical to X-ray SED.** Figure 3 shows the UV and optical SED of the transient for epochs where the quasi-simultaneous Swift UV and OGLE I-band observations are available, along with the best-fitting blackbody curves. The host UV flux is estimated from the host SED fitting (see "Methods": UV to optical SED analysis) and subtracted from the observed emission. The blackbody fitting

results indicate that the temperature appears roughly constant around $T \simeq 16{,}000$ K for the first ~40 d after the optical discovery and then rises to $T \simeq 25{,}000$ K over the next ~30 d, while optical emission re-brightens. Fitting to only the Swift UV data shows that the temperature does not to increase further for the rest of the epochs (Supplementary Fig. 4). On the other hand, the radius remains at ~$2 \times 10^{15}$ cm for the first ~40 d, on the high end of the radius range observed for TDEs[13], after which it decreases by a factor of 3. Also shown in Fig. 3 is the quasi-simultaneous X-ray spectrum observed with XMM-Newton at ~140 d and the best-fitting blackbody model. It can be clearly seen that while the X-ray temperature is an order of magnitude higher, it is not enough to explain the observed UV/optical emission, suggesting physically distinct emission components at the two bands arising probably from different locations. The blackbody radius inferred from the XMM observations is $r_{\mathrm{bb}} \sim 10^{12}$ cm, comparable to the Schwarzschild radius ($R_{\mathrm{s}} = 2GM/c^2$) for a black hole mass of $1.6 \times 10^6\, M_{\odot}$ (ref. [21] and Supplementary Note 1). This suggests that the soft X-ray emission originates from a compact accretion disk. The origin for the UV/optical emission is not clear yet and will be discussed in the next section.

The evolution of the X-ray luminosity with respect to UV/optical luminosity is displayed in Fig. 4 (red symbols). In comparison with the X-ray brightening in other optical TDEs, the evolution for OGLE16aaa presents a sharp increase at ~150 days by about two orders of magnitude within only ~2 weeks. The same trend can be seen from the X-ray luminosity evolution. It is noteworthy that due to the lack of enough data points to verify the actual rise time, it cannot be determined whether the rise to peak time of the X-ray emission for other optical TDEs is as dramatic as OGLE16aaa (Supplementary Note 3 and Fig. 5).

## Discussion

OGLE16aaa is only the seventh optically discovered TDE with a detection of bright X-ray emission ($L_{\mathrm{X}}/L_{\mathrm{opt}} \sim 1$ close to the X-ray

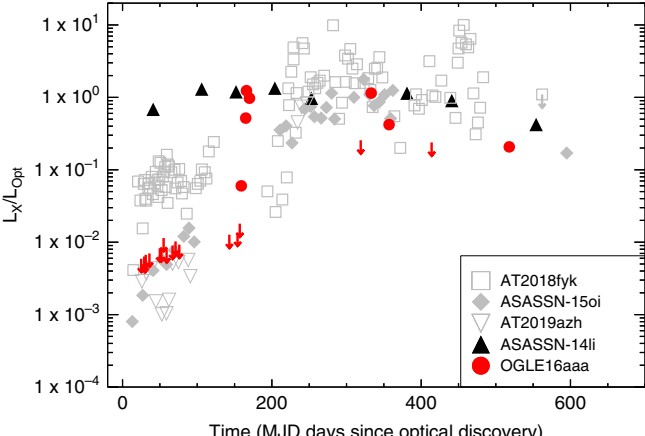

**Fig. 4 Comparison of OGLE16aaa with other TDEs in the evolution of optical to X-ray luminosity ratio.** The optical luminosity refers to the integrated blackbody luminosity from the SED fittings to the Swift UVOT photometry. For epochs where only UV observation at one band is available ($t = [300, 490]$ days), we extrapolated the UV luminosity to the blackbody luminosity by assuming a constant temperature evolution since $t = 150$ days. For the X-ray non-detections, the corresponding $3\sigma$ upper limits are given. Red symbols represent the luminosity ratios of OGLE16aaa at different epochs. For comparison, we also plotted the ratios for other X-ray bright TDEs from[5,6,39], as noted in the legend. ASASSN-14li[17] is the only TDE that shows nearly constant optical to X-ray luminosity ratios since discovery. Although OGLE16aaa appears to follow the evolution trend as other TDEs, the time of rise to peak is distinct.

peak) by Swift and XMM-Newton within a few months since its discovery, and the first source that has a resolved rise-to-peak in both X-ray and optical bands[5,6,17,18]. The X-ray emission exhibits a delayed brightening roughly ~140 days with respect to the optical emission which is also unique among optically discovered TDEs. Many recent numerical studies have shown that the infalling stellar debris stream will undergo self-intersections as a consequence of relativistic apsidal precession[14,29,30], where optical/UV emission could be produced because of shock heating. Following the stream self-interactions, the debris spreads inward and gradually circularizes to form an accretion disk on the free-fall timescale. Within this picture, there will be a time delay between the debris self-crossing and onset of disk formation, possibly explaining the observed delay of the X-ray emission in OGLE16aaa. However, recent simulations of realistic disk formation[31] suggest that the shock heating rate of the initial self-intersections near the apocenter radius might be much weaker than that required to power the optical emission (~$10^{44}$ ergs$^{-1}$); hence, it appears not enough to account for observations. Although the simulations show that self-intersections taking place close to pericenter can produce secondary shocks with high enough heating rate, in this case the debris has reached a significant level of circularization, leading to rapid formation of disk, which seems contrary with the early non-detection of X-ray emission. Therefore, the scenario that the late time X-ray brightening in OGLE16aaa is due to delayed onset of disk formation seems disfavored[32,33].

It has been proposed[16] that if the majority of falling-back debris becomes unbound in a dense outflow, the X-ray radiation from the inner accretion disk will be initially blocked, and may escape at later times as the density and opacity of the expanding outflow decreases[6]. In the model, efficient circularization of the returning debris is assumed, resulting in rapid onset of disk accretion. The timescale for the ionization breakout of X-ray radiation is found about several months for a black hole of $M_{\mathrm{BH}} \sim 10^6\, M_{\odot}$[16], in agreement with the observed time delay of X-ray brightening for OGLE16aaa. In this case, the reprocessing by irradiated ejecta can produce the bright optical emission, for which the radiative efficiency is high enough to naturally explain the bolometric output of most TDEs. The optical emission at early times for OGLE16aaa is likely due to the reprocessing of the X-ray radiation. This is further supported by the ratio of X-ray to optical luminosities (Fig. 4) that is very close to one at the peak. However, if the rapid X-ray brightening is due to the ionization breakout of disk emission, the reprocessing scenario alone cannot explain the late time X-ray evolution either, which is characterized by multiple flux dips and flares.

Alternatively, if the reprocessing layer is moderately patchy, the Keplerian motion could be invoked to explain the unusual X-ray variability observed in OGLE16aaa due to changes in absorption along the line of sight[18]. We can estimate the crossing time for an intervening gaseous material orbiting outside the X-ray source as

$$t_{\mathrm{cross}} = 0.7 \left( \frac{r_{\mathrm{orb}}}{\mathrm{light-day}} \right)^{3/2} M_6^{-1/2} \arcsin\left( \frac{r_{\mathrm{abs}}}{r_{\mathrm{orb}}} \right) \mathrm{yr}, \qquad (1)$$

where $r_{\mathrm{orb}}$ is the orbital radius of absorber, $M_6$ is the BH mass in units of $10^6\, M_{\odot}$, and $r_{\mathrm{abs}}$ is the projected size of the moving absorption gas that is assumed comparable to the orbital radius[34]. Assuming that the distance of the intervening gas is the same as the radius of the photosphere for the UV/optical emission, which is ~$1$–$2 \times 10^{15}$ cm (~0.4–0.8 light day) from the blackbody fittings, and a black hole mass of $10^{6.2}\, M_{\odot}$[21], we obtained a crossing time of ~50–140 days. This is comparable to the duration of X-ray non-detections ($t \sim 140$ days, the period between the optical peak and first appearance of X-ray emission), as well as the time

**Table 3 SMBHB model parameters for OGLE16aaa and J1201 + 3003.**

| Parameter | OGLE16aaa | J1201 + 3003 |
|---|---|---|
| BH mass ($M_\odot$) | $10^6$ | $10^7$ |
| Eccentricity $e$ | 0.4 [0.4, 0.6] | 0.3 [0.1, 0.5] |
| Penetration factor $\beta$ | 4.5 [3.0, 6.0] | 1.3 [1.3, 1.6] |
| Mass ratio $q$ | 0.25 [0.05, 0.9] | 0.08 [0.04, 0.09] |
| Orbital period $T_{orb}$ (days) | 150 [140, 160] | 150 [140, 160] |
| Initial phase $\phi$ | $1.7\pi$ | $1.5\pi$ |

interval of following X-ray flares (Fig. 1b). However, as the rise to peak time is short (~10 days), such a scenario requires extreme condition such as sharp column density transitions near the gap of the intervening gas and so partial covering of the X-ray source to accord with the almost no spectral changes between the low and high X-ray flux states. In addition, the obscuring gas is required to be dense enough to remain opaque for more than 1 year to block X-ray source where the reprocessing efficiency is expected high. In contrast, the optical emission displays an extended plateau between 150 and 300 days, with a flux comparable to the host emission observed in the pre-flare phase, suggesting that reprocessing may be less efficient. The lack of continuous UV observations prevents further investigation on how the UV emission evolved at this epoch, which is crucial to constrain the obscuration scenario. Given the limited dataset, we cannot fully exclude the possibility of that the X-ray behavior is due to the presence of the variable absorption, as proposed for the TDE AT2019ehz[18].

Although it is rarely seen in standard TDEs, the strong flux dip superposed on the overall decline appears to be consistent with the model prediction of tidal disruption by a SMBHB system, where the presence of a secondary perturber can cause gaps in the light curve[35–37]. Such a characteristic flux interruption in the light curve has been observed in SDSS J120136.02 + 300305.5 (J1201 + 3003), the first candidate TDE by a SMBHB in a quiescent galaxy [9,38]. We test this possibility by using the same model proposed for J1201 + 3003[9] with $M_{BH} = 10^6 M_\odot$ and $\theta = 0.3\pi$, where $\theta$ is the inclination angle between the orbital plane of SMBHB and disrupted star. We find that the overall X-ray light curve for OGLE16aaa can be reproduced by the model of tidal disruption by a SMBHB. The fitting results are summarized in Table 3 and shown in Fig. 5a as gray dot-dashed lines. Despite the high number of free parameters, in comparison with the fitting results for J1201 + 3003 (Table 3), the best-fit of the SMBHB model for OGLE16aaa suggests relatively large eccentricities ($e \sim$ 0.4–0.6) and penetration factors ($\beta \sim$ 3–6), whereas the orbital period of SMBHB is similar with $T_{orb} \sim$ 150 days. The constraints on mass ratio are quite uncertain. Both major and minor merger in the models are consistent with the observation ($q \sim$ 0.05–0.9). As no clear tidal features are observed in the optical imaging[21], the minor merger like that inferred in J1201 + 3003 may be more favored.

It is noteworthy that although the X-ray spectra for both objects are extremely soft without emission at energies above 2 keV, a single thermal blackbody model is not sufficient to describe the data, requiring an additional blackbody component (Supplementary Note 4 and Supplementary Fig. 6). We argue that such an excess component is unlikely related to the disk emission of secondary BH in the SMBHB scenario. Given the BH mass of $10^5$–$10^6 M_\odot$ for the secondary (Table 3), the best-fit blackbody temperature is much higher than that expected from standard disk model[12], especially for J1201 + 3003. In addition, as the separation of two SMBHs in our model configuration is relatively

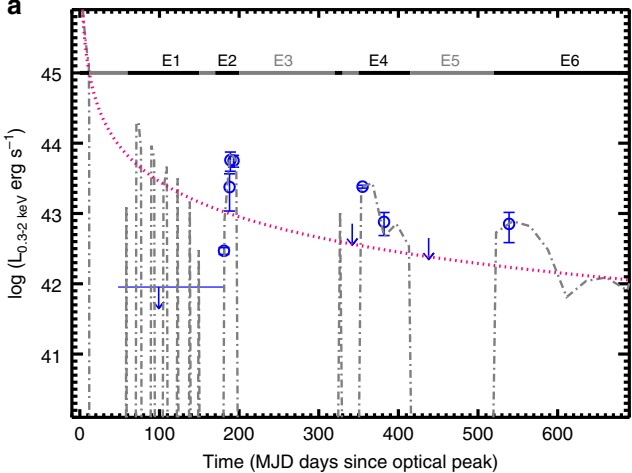

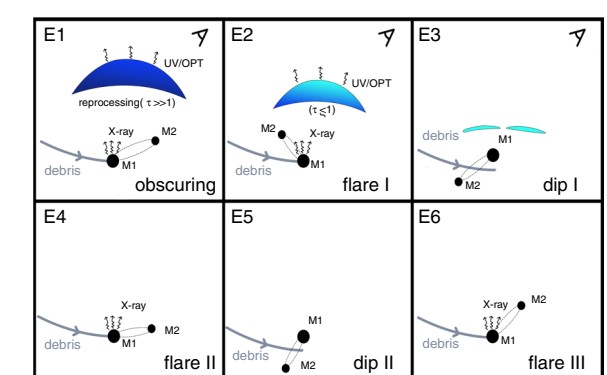

**Fig. 5 X-ray luminosity evolution of OGLE16aaa modeled with the tidal disruption by binary black hole. a** The observed 0.3–2 keV luminosity of OGLE16aaa at different epochs, shown with blue symbols. Error bars correspond to $1\sigma$ uncertainties, as that shown in Fig. 1. The $3\sigma$ upper limits are adopted for X-ray non-detections and shown with downward pointing arrows. Simulated light curve is shown in gray dot-dashed line, for the tidal disruption by binary black hole in the observer frame for OGLE16aaa (Table 3). As there is a time difference between the fallback particles and radiation in simulations, which is 40 days from the best-fit, the observed light curve is shifted rightward to match the simulated one. The magenta-red dotted line represents the canonical $t^{-5/3}$ decay law. Observational data are from Table 1. The horizontal thick lines represent different stages of X-ray evolution, as illustrated in **b**. **b** A schematic illustration of the evolution of X-ray emission. E1: The X-ray emission from the inner accretion disk of primary BH is initially obscured by a thick reprocessing layer, by which the X-ray radiation is absorbed and re-emitted at UV/optical wavelengths; E2: Ionization radiation results in a change in the opacity at later times, and direct escape of X-ray radiation (flare I). The reprocessing layer recedes inwards as $\tau \lesssim 1$; E3: Due to the perturbation of secondary BH, the infalling stream debris misses the accretion radius, causing interruptions in the light curve (dip I); E4: The accretion continues and X-ray flare occurs (flare II); E5: The infalling debris begins to miss again (dip II); E6: Beginning of another stage with accretion (flare III).

large compared with the semi-major axis of the most bound debris, most of materials will be accreted by the primary BH and the contribution from secondary to the X-ray emission is low. In fact, such an additional X-ray emission appears to be ubiquitous in the X-ray spectra of optical TDEs presented in Fig. 4, and is found to vary in flux similar to the primary blackbody component[39,40]. It is likely that the extra component originates from a transient corona that is synchronized with the formation of accretion disk (Supplementary Note 5 and Supplementary

Fig. 7). Accurate modeling the evolution of the extra X-ray emission, beyond the scope of this paper, is necessary to understand its origin.

If the interpretation of the SMBHB is correct, we note that the best-fit model predicts an episodic accretion for a period of ~90 days after the first interruption, producing an X-ray luminosity of ~$10^{43}$–$10^{44}$ ergs$^{-1}$, which is at odds with the Swift non-detections at that period from either individual or combined observations. The observed properties for OGLE16aaa may be broadly consistent with the reprocessing model[16]. The stream collisions lead to the rapid formation of the accretion disk, where the circularization process is efficient likely due to the presence of a secondary BH[36]. The early X-ray non-detections could be attributed to the obscuration by a dense column of gas from unbound outflow, where the radiation heating produces the UV/optical emission. The ionization breakout could allow the escape of X-ray photons at later times, yielding a delayed X-ray emission for which the subsequent evolution is dominated by the discrete accretion in the SMBHB system. We note that the properties of ASASSN 14-li, the only TDE that is both X-ray and optically luminous since discovery[17], can be unified by the reprocessing scenario if it is viewed along the direction with lower density of the ejecta and so does the negligible time for ionization breakout.

It is interesting to note that the optical I-band light curve of OGLE16aaa displays a second peak (rebrightening) around 30 days after the initial peak. During the rebrightening phase, the source also exhibits a possible variability in the UV bands. Although the nature of the rebrightening is under debate, ref.[21] has proposed the possibility of a binary BH on a tight orbit to explain the optical variability in OGLE16aaa. This is reminiscent of the TDE candidate ASASSN-15lh, for which a similar rebrightening in its optical/UV light curve has been observed and could be explained with a model of SMBHB with extreme mass ratio ($q = 0.005$)[41]. Strictly speaking, the SMBHB interpretation for ASASSN-15lh is qualitative, as the simulations show only the evolution in the accretion rate for the secondary BH with a mass of $5 \times 10^5$ M$_\odot$. In this case, the expected luminosity from Eddington accretion is by an order of magnitude less that observed UV/optical luminosity (~$10^{45}$ ergs$^{-1}$), making it unlikely that the UV/optical emission originates directly from the accretion disk. The rebrightening can alternatively be explained as reprocessing of X-ray radiation into UV/optical emission by delayed disk accretion[42] onto a single SMBH, or ionization breakout due to a sudden change in the ejecta opacity[43]. For the latter case, as the model is only sensitive to the UV emission, it seems difficult to explain the optical plateau phase at longer wavelengths[42]. Hence, the ionization breakout is impossible to account for the optical rebrightening in OGLE16aaa and the reprocessing of accretion luminosity remains the most likely scenario. As the rebrightening phase appears relatively short-lived, the reprocessing scenario requires fluctuations in the mass accretion rate. This is not inconsistent with the SMBHB model because of the presence of many accretion islands in the early times, as shown in Fig. 5a. Unfortunately, as we lack enough data points after the rebrightening phase in the I-band light-curve, it cannot be determined whether the optical emission will display further variability or decay smoothly, which is crucial to test the SMBHB scenario.

We conclude that the overall properties of OGLE16aaa could be accounted for by the stellar tidal disruption by a candidate SMBHB. The delayed brightening in the X-ray emission as well as the multiple flux dips during the decay phase are in agreement with a SMBHB model with a mass of $10^6$ M$_\odot$ for the primary BH, mass ratio of 0.25 and orbital period of 150 days (Table 3). In comparison with the prediction by the SMBHB model, the X-ray non-detections in the early phase could be attributed to the obscuration by a dense column of gas, perhaps from unbound outflow. This implies that the reprocessing may be a viable mechanism to explain the UV/optical emission at the same epoch. Ionization breakout allows for the escape of X-ray photons, resulting in the detectable X-ray emission at later times ($t \gtrsim 140$ days). A schematic illustration of this process is shown in Fig. 5b.

If our interpretation with the SMBHB model is correct, OGLE16aaa could be the second TDE candidate with a SMBHB at its core revealed in the X-rays. Upon final coalescence, SMBHB systems like the one in OGLE16aaa (and SDSS J1201 + 3003) are prime sources for future space-based GW missions like Laser Interferometer Space Antenna[44]. Note that given the estimated GW inspiral time of $t_{gw} \sim 1.9 \times 10^7$ years[45], it would be challenging to detect the GWs from such SMBHB system in its current state of evolution. In synergy with Large Synoptic Survey Telescope[46], the future X-ray sky surveys such as extended Roentgen Survey with an Imaging Telescope Array[47] and Einstein Probe[48] are expected to detect similar TDEs more than one hundred[49], providing a powerful tool for studying the physics on how the stellar debris evolves after disruption, and searching for promising candidates of milliparsec SMBHBs that are still poorly explored.

## Methods

**X-ray data reduction**. OGLE16aaa has two XMM-Newton observations, which were performed on Jun 2016 and Nov 2016 (XMM1 and XMM2), with an exposure of 15ks and 36 ks, respectively. The XMM-Newton data were reprocessed with the Science Analysis Software version 16.0.0, using the calibration files that are available up to December 2018. The time intervals of high background events were excluded by inspecting the light curves in the energy band above 12 keV where the count rates for source are low. Detailed spectral analysis was performed only on the data taken with PN[50], as it has a much higher sensitivity, while the MOS[51] data have been used to check for consistency when necessary. The source spectra were extracted within a circular region centered at the source optical coordinate, with a radius of 35″. Background spectra were extracted from clean regions on the same chip using four identical circular regions to that of source. We grouped the spectra to have at least 5 counts in each energy bin so as to adopt the $\chi^2$ statistic during the process of spectral fitting. Since no hard X-rays are detected at energies above 2 keV, we performed spectral fittings in the 0.3–2 keV range using XSPEC (version 12). All statistical errors given correspond to the 90% confidence intervals for one interesting parameter ($\Delta\chi^2 = 2.076$), unless stated otherwise.

All Swift observational data were retrieved from the HEASARC data archive. Details of the Swift observations can be found in Supplementary Table 1. The calibration files are taken from that released on 13 November 2017. The XRT was operated in Photon Counting mode. We reprocessed the XRT event files with the task xrtpipeline (version 0.13.2). For each Swift-XRT observation, we used XSELECT which is part of HEASoft (FTOOLS 6.19) to extract the source spectrum with a circular region of 40″ radius. Background spectrum was extracted from an annulus region centered on the source position, with an inner radius of 60″ and outer radius of 110″, respectively. For the data taken in first 16 epochs (from 19 January to 8 June 2016), no X-ray signal was observed at the location of OGLE16aaa in either individual or stacked images. The corresponding $3\sigma$ upper limit was estimated from the background region with the spectral extraction task in the HEASoft package. The source is not detected either on 17 November 2016 and the epoch between 19 and 23 February 2017. The X-ray upper limits and the detections from Swift observations are listed in Table 1. The count rates were converted into flux using the WebPIMMS tool, assuming a blackbody model with temperature of $kT_{BB} = 60$ eV modified by a Galactic HI column density of $N_H = 2.71 \times 10^{20}$ cm$^{-2}$.

**UV and optical data reduction**. UV imaging data of OGLE16aaa were obtained with the Swift UVOT instrument in three filters: UVW1 (2600 Å), UVM2 (2246 Å), and UVW2 (1928 Å). For the most recent observations performed on 12 January 2020, imaging data in three optical filters, U (3465 Å), B (4392 Å), and V (5468 Å), are also available, which allow for better determining the host emission. We used the UVOT software task UVOTSOURCE to extract the source counts from a region with radius of 5″. The background is chosen from a source-free sky region with a radius of 40″. The UVOT count rates were then converted into AB magnitudes which are presented in Supplementary Table 1. The OGLE-IV I-band photometric data are publicly available and can be found in http://ogle.astrouw.edu.pl/ogle4/transients/. The host contribution to the I-band flux was subtracted using the pre-discovery images as templates. For the UVOT data, as no good pre-discovery reference images are available and the photometric errors from the most

recent observations are large, we used the model galaxy spectrum from the SED fit to generate synthetic magnitudes at these wavelengths. We obtained the pre-flare measurements of OGLE16aaa from the NED database (https://ned.ipac.caltech.edu), including near-UV (NUV) and far-UV (FUV) photometry from GALEX, optical b_J photometry from the APM survey, near-IR JHK$_s$ photometry from the 2MASS and mid-IR photometry at 3.6 and 4.5 μm from WISE. In addition, we also used the UVOT photometry from the most recent observations taken nearly 1480 days since optical outburst, for which the emission is likely dominated by the host component. All the host photometric data are presented in Supplementary Table 2. We fitted the above photometry of the host using the code FAST and the best-fitting model is shown in Supplementary Fig. 1 (and Supplementary Note 1). In the AB system, we obtained the host magnitudes of UVW1 = 20.52 ± 0.18 mag, UVM2 = 20.75 ± 0.21 mag and UVW2 = 21.03 ± 0.18 mag, which were then subtracted from the UVOT measurements to obtain the transient photometry (and the errors on host flux were propagated). In addition, all flux densities have been corrected for the Galactic extinction of $E(B − V) = 0.018$ mag. It is noteworthy that we did not corrected for an internal extinction by host, as the uncertainties on the best-fit extinction given by FAST are large ($A_V = 1.0^{+1.65}_{-0.3}$, 68% confidence intervals).

**UV to optical SED analysis**. We fitted a blackbody model ($B_\nu = \frac{2h\nu^3}{c^2}\frac{1}{e^{h\nu/kT}-1}$) to the host-subtracted, extinction-corrected photometric data from the Swift UVOT observations, to put constrains on the luminosity, temperature and radius evolution of UV and optical emission. Uncertainties on the above parameters were derived using Monte Carlo simulations, in which the observed fluxes were randomly perturbed with amplitude by assuming Gaussian noise according to the photometric errors. This procedure was repeated 1000 times. The error bars on each parameter were then derived from the 16th and 84th percentiles of the distribution of the corresponding values obtained in the simulations. Using the best-fit temperature and rest-frame monochromatic UV luminosity at each epoch, we estimated the blackbody radius from $L_\nu = \pi B_\nu \times 4\pi R^2_{BB}$ and took blackbody luminosity from $L_{BB} = \sigma T^4 \times 4\pi R^2_{BB}$ as the integrated luminosity of UV and optical emission. The evolution of blackbody luminosity, temperature and radius are presented in the Supplementary Fig. 4.

## Data availability

Source data for the observations taken with XMM-Newton and Swift are available through the HEASARC online archive services (https://heasarc.gsfc.nasa.gov/docs/archive.html). Optical imaging data at I-band are publicly available through the website of the OGLE-IV Transient Detection System (http://ogle.astrouw.edu.pl/ogle4/transients/2017a/transients.html). The authors can provide other data that support the findings of this study upon request.

## Code availability

The Science Analysis Software used to reduce the XMM-Newton data is publicly available at https://www.cosmos.esa.int/web/xmm-newton/download-an. The X-ray analysis softwares, XSPEC and XSELECT, are part of HEASoft, which can be found at https://heasarc.gsfc.nasa.gov/docs/software/heasoft. The Swift data analysis software (xrtpipeline and UVOTSOURCE) can be found at https://swift.gsfc.nasa.gov/analysis/. WebPIMMS tool is available at https://heasarc.gsfc.nasa.gov/cgi-bin/Tools/w3pimms/w3pimms.pl. The code used to model the UV-to-MIR SED is accessible through github (https://github.com/jamesaird/FAST). The other codes that support the plots within this article are available from the authors upon reasonable request.

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

## Acknowledgements

This research made use of the HEASARC online data archive services, supported by NASA/GSFC. We thank the Swift and XMM-Newton observatories, and the OGLE-IV Transient Detection System (OTDS) for making the data available. X.S. thanks Zuozifei Song for producing the schematic diagram in Fig. 5. This work is supported by Chinese NSF through grant numbers 11822301, 11833007, U1731104, and 11421303. S.L. is grateful to the Key International Partnership Program of the Chinese Academy of Sciences (CAS) (number 114A11KYSB20170015) and the Strategic Priority Research Program (Pilot B) "Multi-wavelength gravitational wave universe" of CAS (number XDB23040100).

## Author contributions

X.S. performed the data analysis and prepared the paper. W.Z. and L.D. reduced the XMM-Newton and Swift spectral and photometry data. S.L. performed the X-ray light curve fitting with the SMBHB TDE model. N.J. and L.S. commented the optical and X-ray data analysis, and contributed to the UV-optical spectral energy distribution fitting. T.W., F.L., Z.Y., F.X., and R.S. contributed to the overall interpretation of the results. All the authors joined the discussion at all stages.

## Competing interests

The authors declare no competing interests.
