## [Peer Review File · Nature Communications]

REVIEWER COMMENTS

Reviewer #1 (Remarks to the Author):

I would like to reveal my identity to the Authors as Lukasz Wyrzykowski, who discovered the OGLE16aaa. I have read the paper of Shu et al. with high interest, as the possibility of discovering another binary SMBH is very exciting. I am in favour of this article to be published in Nature Communications after my comments below are addressed.

1. The strongest evidence for the binarity of the SMBH here is the presence of the dips in the X-ray light curve, which have not been seen in many other X-ray TDEs. However, the optical variability of TDE optical light curves has been seen in recent TDEs, e.g. AT2018fyk, and is not attributed to the binarity of the SMBH. The claim in my discovery paper Wyrzykowski 2017 is not valid anymore.
2. I don't agree with the statement that OGLE16aaa shows the most dramatic rise in the X-ray L_x/L_{opt} plot (Fig.4). All other examples show simply lack data to verify the rise time. This seems like an observer's luck here.
3. Why is Fig.4 not showing the dips in X-ray luminosity at later times? Where is the data at epochs around 300days? Showing the dips in that plot would make OGLE16aaa stand out among other TDEs, as this is the unique property of this TDE.
4. I would like to see more discussion on the comparison between OGLE16aaa and AT2018fyk, ASASSN-15oi and AT2019azh, as the X-ray delay is present in all these cases. Why the other three are not due to bin-SMBH?
5. The additional blackbody component seen in the X-ray spectrum is not explained enough, in my view. How does it connect to a potential scenario of binary SMBH? Note that in AT2018fyk the X-ray spectrum also had to be fitted with an additional component.
6. As the discoverer of the OGLE16aaa event, I would like to request to report the name of the event in the abstract. Also in section 2 (Results) please describe the discovery as:

The optical transient (...) was discovered by the Optical Gravitational Lensing Experiment (OGLE-IV, cite Udalski 2015) and its Transient Detection System (cite Wyrzykowski 2014)

7. I encountered numerous typos in the text, e.g. "rebrigtens", but I hope those will be noted by the editors.

Reviewer #2 (Remarks to the Author):

This paper presents an observational analysis of the previously discovered tidal disruption event (TDE) candidate OGLE16aaa (Wyrzykowski et al. 2017) based on existing and additional data in the optical, UV and X-ray bands. The main new finding consists in the detection of X-rays from the source, with the associated light curve displaying a later peak compared to the optical and UV signals as well as several dips. Several scenarios are proposed to interpret the observations that depend on the mechanism at the origin of optical emission in TDEs and invoke the presence of a binary black hole or a patchy absorber.

The observations made in this work are convincing and represent a significant addition to the literature already available for this object. A clear and complete description of the observational results and the methods used to obtain them is provided. However, I find that the interpretation lacks clarity such that it is difficult to identify the main conclusions made by the authors and the motivation for the physical explanations they favor. To improve this point, I recommend to significantly reorganize Section 3. This feedback is detailed in the major and minor comments below that should be addressed before the manuscript can be considered for publication.

>>>> Major comments (by order of appearance in the text)

Abstract: the authors write that the nature of the X-rays detected from TDEs is "debated". However, most theoretical works agree that this emission originates from the accretion of gas onto the black hole. This part of the text should be revised to clarify this point.

Second paragraph of Section 1: in the first sentence, the authors seem to imply that current observations favor that TDEs emit mostly in the UV to soft X-ray bands, as found in early theoretical works. However, the majority of events have optical emission in excess to what this picture predicted, as correctly pointed out in the rest of the paragraph. This point should be clarified.

First paragraph of Section 2: the authors mention that the I-band light curve displays a re-brightening after the initial decay from the peak. However, this feature is not evident from the black markers in the upper panel of Fig. 1. The time where the optical luminosity starts to increase again should be indicated on the figure along with a reference in the text such that it becomes clearly identifiable.

First paragraph of Section 2: the authors write that they find “evidence for another decline until 140 days”. However, this statement does not seem supported by observations since the I-band light curve does not have data points between $t = 40$ days and $t = 140$ days. This point should be clarified.

Second paragraph of Section 2: for clarity, the dips in the X-ray light curve should be described in this paragraph rather than in the third paragraph of Section 3. In addition, the meaning of the downward pointing arrows present in the lower panel of Fig. 1 should be explained in the text and in the caption of the figure.

Section 3: the main goals of this section is determine the origin of the delayed X-ray emission compared to optical light and the dips in the X-ray light curve. To this end, the authors introduce several mechanisms where the optical emission is produced either by shocks or reprocessing and invoke the presence of a black hole binary or a patchy absorber. However, the main conclusions of the analysis are unclear and I recommend to significantly reorganize the section to improve this point. The text should be modified to clearly present the different proposed scenarios and the ability of each of them to account for observations. The interpretations favored by the authors should be highlighted along with the main reasons for this choice. This revision is the most urgent to make about this section. In addition, more comments below aim at further improving specific aspects of the analysis.

First paragraph of Section 3: the authors mention one possible origin for optical emission that results from shocks while the X-rays are produced by the accretion of the newly-formed disc. Within this picture, they propose that the delay of the X-rays is caused by the time required for the accretion disc to form. However, numerical simulations (e.g. Shiokawa et al. 2015; Bonnerot and Lu 2020) find that a peak of the shock-heating rate of $\sim 10^{44}$ erg/s requires collisions near the black hole that only take place once the assembly of the accretion disc has essentially completed. This seems to disfavor the late onset of the disc as the origin for the delayed X-rays, although the theoretical

understanding of this process is still uncertain. Another perhaps more likely origin comes from the viscous timescale required for the disc to start accreting after it formed. Simulations suggest that it can reach a few fallback times t_0 because most of the gas is located at distances from the black hole much larger than the stellar pericenter. These different options should be discussed in the text when describing the scenario where optical emission is due to shocks.

Third paragraph of Section 3: as explained above, most of this paragraph should be moved to Section 2 for clarity.

Fourth paragraph of Section 3: the term “fully consistent” seems to imply that the presence of a binary black hole is the only possible explanation for the dips seen in the X-rays. Because other possibilities such as that relying on a patchy absorber appear as likely to explain of this feature, these terms should be changed.

Fifth paragraph of Section 3: this paragraph is dedicated to the scenario where the optical photons are produced by re-processing and should not rely on the presence of a black hole binary. As part of the general revision of this section recommended above, I suggest that the authors include the first sentence of this paragraph to the previous one that focuses on the scenario where a binary black hole is assumed.

Fifth paragraph of Section 3: according to Fig. 4, the ratio of X-ray to optical luminosities is very close to one at peak. This value seems to represent a strong evidence for re-processing of the X-rays to be at the origin of the optical photons. If they were instead produced by shocks that are independent on the efficiency of accretion, there would be no reason for these two luminosities to be equal. This argument has for example been used by Van Velzen et al. (2020) when analyzing the X-ray flares seen from AT2019ehz, one of the seventeen TDE candidates detected by ZTF. It should be included in the revised version of the paper.

Sixth paragraph of Section 3: the authors argue that the re-brightening of the optical light curve could be due to the presence of a black hole binary. However, other possibilities have been proposed in the context of ASASSN-15lh that only rely on a single black hole (Leloudas et al. 2016; Margutti et al. 2017). These other interpretations should be included along with potential reasons for favoring the scenario put forward by the authors.

Seventh paragraph of Section 3: the authors propose that the dips in the X-rays is due to the passage of a dense gas cloud along the line of sight. They should explain the calculation made to obtain the

value of $t_{\text{cross}} = 150$ days while the period of interruption can be as short as the exposure time of 1.7 ks used to obtain the X-ray upper limits with Swift.

Last paragraph of Section 3: as part of the revision of this section recommended above, this paragraph should be modified so that it clearly summarizes the main conclusions made by the authors regarding the interpretation of the observational data.

Last paragraph of Section 3: the authors write that the origin of the delayed X-rays from a delayed disc assembly or accretion is disfavored based on the temporal evolution of the optical and X-ray signals but this claim is not justified in the paper. This statement should be motivated in the earlier parts of this section.

>>>> Minor comments (by order of appearance in the text)

Abstract: there appears to be words missing in the sentence containing “and promising prospect”. This should be fixed.

Second paragraph of Section 1: there is a typo where “the first a few months” is written instead of “the first few months”. This should be corrected.

Second paragraph of Section 2: there is a typo where “reaching to an X-ray peak” is written instead of “reaching an X-ray peak”. This should be fixed.

Third paragraph of Section 2: there is a typo where “lack for emission” is written instead of “lack emission”. This should be fixed.

Fifth paragraph of Section 3: there are words missing in the sentence containing “X-rays is found about”. This should be corrected.

Note 1 of the supplementary information: a reference appears to be missing that has been replaced by a question mark. This should be fixed.

Caption of the supplementary Figure 4: the description of the top and bottom panels is inconsistent with the annotations in the plots referring to OGLE16aaa and J1201. This should be fixed.

Reviewer #3 (Remarks to the Author):

In their paper, "An X-ray flare from the stellar tidal disruption by a candidate supermassive black hole binary," the authors present an analysis of the optical/X-ray TDE candidate OGLE16aaa, and they investigate the possibility that the variability observed in the X-ray flux is due to the presence of a binary companion. They also consider the alternative scenario that the dips in the X-ray lightcurve are due to changes in the absorption along the line of sight.

Overall, I think the analysis carried out in this paper is interesting, and the interpretations seem broadly consistent with the observations. I have some comments/questions that I would like to see addressed before the paper is accepted for publication.

1) By-eye, the fit to the X-ray data in Figure 3 seems to over-represent the true flux at most wavebands. Is there a reason for this, i.e., is the fit being skewed by the one, apparent outlier point above the curve around ~ 0.8 keV? More specifically, it seems as though reducing the temperature by a modest factor would provide a much better fit for the data in the range between ~ 0.2 and 0.8 keV. The authors should verify their fit and/or provide an explanation for this apparent by-eye discrepancy.

2) The authors state that the I-band flux re-brightened after roughly a month. Investigating Figure 1, it appears as though there is tentative evidence for the UVW2 and UVM2 bands displaying a similar brightening. Is this statistically significant? If so, the authors should simply comment on this.

3) Overall I agree with the authors that the X-ray luminosity is fairly erratic over the time that the system was observed, and in particular the extremely short rise time is uncharacteristic of a TDE. However, have the authors attempted to fit the overall trend in the declining X-ray luminosity with a $t^{-5/3}$ power-law, similar to the optical data? If there were simply too much obscuring material at early times that prevented detection, the existence of such a power-law decline would hint at the validity of this argument (i.e., that it is the same underlying source powering the emission, and the X-rays were just obscured at early times).

4) The authors state that the time to peak in the X-rays is given by the vertical, dashed, gray line in the bottom panel of Figure 1, which is constrained by the 4 very tightly spaced points around a time of ~ 140 days and in particular the 3 Swift X-ray observations. However, the difference in X-ray luminosity between the two points near the peak is extremely small, and then there appear to be no further observations between the last Swift point around 140 days and the non-detection near day 300. My question is therefore: how confident can the authors be that the true peak in the X-rays was not significantly later than the claimed peak, given the lack of observations? It would help if the authors could constrain the average decline per point 3 above, which would potentially constrain the location of the true peak if one assumes a $t^{-5/3}$ decline.

5) In Figure 2, the authors should show at least one of the results of fitting the additional power-law/blackbody models to the data in addition to the ratio of the data to the model (i.e., similar to the top panel of this figure; this could also conceivably be achieved by simply adding another line that includes the power-law/blackbody component to the top panel of this figure).

6) The authors propose that the declines in X-ray flux are consistent with the predictions from the scenario of the disruption of a star by a SMBH binary. However, the theoretical arguments behind this prediction are that the X-ray luminosity scales with the fallback rate, which is temporarily interrupted as the secondary perturbs the orbits of the infalling debris. It is therefore not obvious to me as to why *only* the X-ray emission would be stunted/interrupted in this model; why wouldn't the UV/optical also show some signs of interruption if the flow and the resulting accretion luminosity are driven by the returning debris? Some explanation along these lines should be provided.

7) In the binary disruption scenario, in addition to the accretion rate onto the primary being reduced, there can also be an increase in the accretion rate onto the secondary as it strongly interacts with the stream of debris produced by the disruption; based on the models of Ricarte et al. (2016) (<https://ui.adsabs.harvard.edu/abs/2016MNRAS.458.1712R/abstract>) and Coughlin et al. (2017) (<https://ui.adsabs.harvard.edu/abs/2017MNRAS.465.3840C/abstract>), this becomes increasingly likely as the mass ratio of the binary nears unity, and certainly seems like it should be possible for the best-fit mass ratio of $q=0.25$ adopted by the authors. Is this additional reservoir of accretion energy accounted for by the model employed by the authors? The authors should at least

mention the possibility that a contribution to the X-ray flux (and presumably other wavelengths) could be provided by the accreting secondary.

8) It would be useful if the authors included a diagram that indicates the qualitative features of their favored scenario, i.e., an envelope that reprocesses/obscures the X-rays to some time, and a binary companion that induces variability in the fallback rate onto the primary.

9) In their conclusions, the authors should note the gravitational-wave inspiral time of the SMBH binary used in their model for the fallback.

Dear Referees,

We are grateful for the detailed reviews and constructive comments on our manuscript. We describe below our replies (in blue) attempting to address each comment raised in the reviews, followed by relevant changes to the revised manuscript (highlighted in bold font).

**Xinwen Shu
On behalf of co-authors**

Reviewer #1 (Remarks to the Author):

I would like to reveal my identity to the Authors as Lukasz Wyrzykowski, who discovered the OGLE16aaa. I have read the paper of Shu et al. with high interest, as the possibility of discovering another binary SMBH is very exciting. I am in favour of this article to be published in Nature Communications after my comments below are addressed.

1. The strongest evidence for the binarity of the SMBH here is the presence of the dips in the X-ray light curve, which have not been seen in many other X-ray TDEs. However, the optical variability of TDE optical light curves has been seen in recent TDEs, e.g. AT2018fyk, and is not attributed to the binarity of the SMBH. The claim in my discovery paper Wyrzykowski 2017 is not valid anymore.

A: We agree with that the dips in the X-ray light curve are of a unique property for OGLE16aaa, which could be explained by the presence of SMBH binary (SMBHB) when compared with the expectation of simulations. However, after a census of the literature, we found that the optical re-brightening following the initial peak is still an unusual feature for the source. Specifically, the optical re-brightening is flare-like and appears to be short-lived with a duration of less than one month. This is remarkably different from AT2018fyk and ASASSN-15lh, where the optical re-brightening is much more pronounced and exhibits a plateau phase lasting for $t \sim > 100$ days. If the optical re-brightening is due to the reprocessing that is powered by accretion onto SMBH, as proposed by Leloudas et al. (2016, NA, 1, 2) for ASASSN-15lh and Wevers et al. (2019, MN, 488, 4816) for AT2018fyk, the shorter timescale for the flare-like re-brightening in OGLE16aaa suggests possibly a variability in the accretion rate, perhaps due to the perturbation by the presence of a secondary BH. On the other hand, Coughlin et al. (2018, MN, 474, 3857) proposed an alternative interpretation for the re-brightening observed in ASASSN-15lh that could be produced by the tidal disruption by an extreme mass ratio SMBHB ($q=0.005$), though only qualitative comparison of the results between simulations and observations were presented. Therefore, we think that the relatively short-lived

optical flare in OGLE16aaa is still a unique characteristic to hint the binarity of the SMBH. Future observations of similar flares in other TDEs would be interesting, and be critical to test the possibility of such association. In the revision, we have updated the discussion on the possible origin of optical re-brightening in the Page 7, third paragraph.

2. I don't agree with the statement that OGLE16aaa shows the most dramatic rise in the X-ray L_x/L_{opt} plot (Fig.4). All other examples show simply lack data to verify the rise time. This seems like an observer's luck here.

Figure A1: The X-ray light curve for the optical TDE AT2018fyk, ASASSN-15oi, and AT2019azh, as compared with OGLE16aaa.

A: In addition to OGLE16aaa (the source of our interest), we retrieved and processed all the X-ray data observed with Swift/XRT up to 2020 June, for the optical TDEs AT2018fyk, ASASSN-15oi, and AT2019azh. The X-ray luminosity evolution as a function of time is shown in Figure A1. The referee is right that because of lacking enough data, we cannot determine whether the rise to peak time for other TDEs is as abrupt as OGLE16aaa. In fact, only in ASASSN-15oi the X-ray brightening appears relatively smooth and less dramatic than that of OGLE16aaa. In order to clarify this point, we have rephrased the statement in describing the X-ray evolution in the revision, at the end of Section 2. The comparison of X-ray luminosity evolution is now included in the Supplementary information (Supplementary Note 3 and Figure 5).

3. Why is Fig.4 not showing the dips in X-ray luminosity at later times? Where is the data at epochs around 300days? Showing the dips in that plot would make OGLE16aaa stand out among other TDEs, as this is the unique property of this TDE.

A: We thank the referee to point this out. This is because we required observations at three UV bands (UUV1, UVM1 and UUV2) to yield meaningful measurements of UV/optical blackbody luminosity from model fittings. As can be seen in the Supplementary Table 1, only observation at one UV band is available between $t=300$ and 490 days, the period covering the X-ray dips. So we did not show the corresponding ratio of X-ray to optical luminosities. In order to obtain the data for the X-ray dips where only single band UV data are available, we first calculated the ratios of UV to total blackbody luminosities that are measurable. By assuming the blackbody temperature does not change since $t=150$ days, we then scaled the UV photometry to estimate the corresponding blackbody luminosity. Note that if extrapolating the blackbody luminosity evolution by fitting a $t^{-5/3}$ law (Supplementary Figure 4, top panel) yields similar results. The resulting ratios of X-ray to optical (blackbody) luminosities are now included in Figure 4, with a note in the caption to explain how it is estimated. In addition, we have updated Figure 4 to include more Swift data for the source AT2018fyk than that presented in Wevers et al. (2019), where only Swift data up to $t=120$ days since discovery are shown, in order to cover the ratios of X-ray to optical luminosities close to the X-ray peak.

4. I would like to see more discussion on the comparison between OGLE16aaa and AT2018fyk, ASASSN-15oi and AT2019azh, as the X-ray delay is present in all these cases. Why the other three are not due to bin-SMBH?

A: As pointed out by the referee, our claim of binarity of the SMBH in OGLE16aaa is mainly based on the erratic X-ray emission, characterized by multiple dips and flares in the lightcurve. Such a characterized X-ray evolution has been investigated in detail with the simulations of tidal disruption by SMBHB (e.g., Liu et al. 2009, ApJ, 706, L133; Ricarte et al. 2016, MN, 458, 1712), where the perturber by the presence of a secondary BH can cause discrete accretion of gaseous debris onto the primary BH, hence flux interruptions in the light curve. Although AT2018fyk, ASASSN-15oi and AT2019azh have shown delayed X-ray brightening, non of them displays clear evidence of interrupted X-ray lightcurve as seen in OGLE16aaa. Therefore, the possibility for the presence of SMBHB in the three sources is not evident (the gaps in the lightcurve is simply due to the lack of observations). More detailed comparison with OGLE16aaa can be found in our reply to the point #3 above, as well as the point #5 below.

5. The additional blackbody component seen in the X-ray spectrum is not explained enough, in my view. How does it connect to a potential scenario of binary SMBH? Note that in AT2018fyk the X-ray spectrum also had to be fitted with an additional component.

A: The referee is right that the additional component in the X-ray spectrum is not well explained, and its origin remains unclear. While the temperature for the primary blackbody component ($kT_{\text{BB}} \sim 50$ keV) agrees with the expectation from the disk accretion onto the BH with a mass of $\sim 10^6 M_{\text{sun}}$, the thermal temperature for the additional component is by a factor of two higher than the primary one. If it relates to the disk emission of secondary BH in the SMBHB scenario, the implied BH mass is lower by at least an order of magnitude (T_{eff} is proportional to $(M_{\text{BH}})^{-1/4}$). This seems inconsistent with our best-fit mass ratio of $q=0.25$, though it is poorly constrained. In fact, in our model configuration where the separation of two SMBHs is relatively large in comparison with the semi-major axis of the most bound debris, most of materials will be accreted by the primary BH, as suggested by simulations (Ricarte et al. 2016). Thus we conclude the possibility that the accretion of debris onto secondary BH contributes to the additional X-ray emission is low.

Figure A2: Blackbody temperature versus photon index from the best-fitting model to the X-ray spectra of optical TDEs. The X-ray TDE SDSS J1201 is also shown for comparison. The X-ray spectra are observed from XMM-Newton which have best S/N. The photon index represents the powerlaw model that is used to describe the additional X-ray component with respect to the primary blackbody emission.

In fact, such an additional X-ray emission component appears to be ubiquitous in the X-ray spectra of optical TDEs (e.g., Gezari et al. 2017; Kara et al. 2018; Wevers et al. 2019; Liu et al. 2020), for which the X-ray luminosities close to peak are bright and comparable to the UV/optical ones. In order to further investigate on the origin of the additional X-ray component, we performed uniform spectral fittings for optical TDEs presented in the Figure 4

of main text, as well as the previously claimed SMBHB TDE candidate SDSS J1201+3003 for comparison, using a blackbody model plus an extra powerlaw component. In Figure A2, we plot the blackbody temperature versus photon index obtained from the best-fittings, while the region enclosed by the green rectangle represents that for typical AGNs (Crummy et al. 2006, MN, 365, 1067). We find a clear diversity of X-ray spectral properties. Four TDEs have a photon index for the extra powerlaw component consistent with that of typical AGNs. However, we argue that this does not necessarily mean the underlying weak AGN emission as origin for the extra component, because its flux appears to vary with time (Kara et al. 2018, MN, 474, 3593; Liu et al. 2019, arXiv:1912.06081). For AT2018fyk, we also find rapid steeping in the photon index (from $\Gamma=2.3$ to $\Gamma=4.1$) within only one year, which is atypical in AGNs. For ASASSN14li and OGLE16aaa, the best-fit yields a photon index for the extra component of $\Gamma>4$, which is extremely rare among AGNs. Only one bona fide AGN of this kind has been reported so far (RX J1302+2747, Sun, Shu & Wang 2013, ApJ, 768, 167). Given its rapid flux and spectral variability, one possible origin for the extra component might be a transient corona that is connected with the formation and evolution of accretion disk. The different photon indices observed could be simply explained by the variety in the physical condition of the coronal region. However, since the formation and evolution of hot corona in TDEs are poorly studied so far, our tentative interpretation for the origin of the additional X-ray component is still speculative. In the revision (Page 7, first paragraph), we have discussed shortly on the possible origins of the additional X-ray spectral component in OGLE16aaa. Detailed comparison with other TDEs has now been presented in the Supplementary Note 5 and Figure 7.

6. As the discoverer of the OGLE16aaa event, I would like to request to report the name of the event in the abstract. Also in section 2 (Results) please describe the discovery as:

The optical transient (...) was discovered by the Optical Gravitational Lensing Experiment (OGLE-IV, cite Udalski 2015) and its Transient Detection System (cite Wyrzykowski 2014)

A: Following the referee's suggestion, we have updated the abstract and the sentence to better describe the discovery of the source.

7. I encountered numerous typos in the text, e.g. "rebrigtens", but I hope those will be noted by the editors.

A: We thank for the referee's careful readings. We have checked through the text in the revision to correct typos as possible as we can (highlighted in bold in the text).

Reviewer #2 (Remarks to the Author):

This paper presents an observational analysis of the previously discovered tidal disruption event (TDE) candidate OGLE16aaa (Wyrzykowski et al. 2017) based on existing and additional data in the optical, UV and X-ray bands. The main new finding consists in the detection of X-rays from the source, with the associated light curve displaying a later peak compared to the optical and UV signals as well as several dips. Several scenarios are proposed to interpret the observations that depend on the mechanism at the origin of optical emission in TDEs and invoke the presence of a binary black hole or a patchy absorber.

The observations made in this work are convincing and represent a significant addition to the literature already available for this object. A clear and complete description of the observational results and the methods used to obtain them is provided. However, I find that the interpretation lacks clarity such that it is difficult to identify the main conclusions made by the authors and the motivation for the physical explanations they favor. To improve this point, I recommend to significantly reorganize Section 3. This feedback is detailed in the major and minor comments below that should be addressed before the manuscript can be considered for publication.

>>>> Major comments (by order of appearance in the text)

Abstract: the authors write that the nature of the X-rays detected from TDEs is “debated”. However, most theoretical works agree that this emission originates from the accretion of gas onto the black hole. This part of the text should be revised to clarify this point.

A: We thank the referee for pointing this out and our description may be a bit misleading. We intended to say the X-ray weakness in most of the optically discovered TDEs from follow-up observations, and the reason for the X-ray weakness is not clear and so far under debate. We have rephrased the sentence to clarify this point.

Second paragraph of Section 1: in the first sentence, the authors seem to imply that current observations favor that TDEs emit mostly in the UV to soft X-ray bands, as found in early theoretical works. However, the majority of events have optical emission in excess to what this picture predicted, as correctly pointed out in the rest of the paragraph. This point should be clarified.

A: We thank the referee for pointing this out. We simply deleted the sentence

“a scenario now favored by a broad range of observations” and updated the references accordingly. Since the discrepancies with the theoretical prediction are mainly resulted from the observations at UV and optical bands, we have reworded the paragraph to better reflect this fact.

First paragraph of Section 2: the authors mention that the I-band light curve displays a re-brightening after the initial decay from the peak. However, this feature is not evident from the black markers in the upper panel of Fig. 1. The time where the optical luminosity starts to increase again should be indicated on the figure along with a reference in the text such that it becomes clearly identifiable.

A: Following the referee’s suggestions, we have now updated the figure and text to show explicitly the time of optical re-brightening. Because there is a gap between $t=15$ days and $t=24$ days, where the flux turns over from $i=19.57$ mag to $i=19.51$ mag, we choose $t=20$ days as the approximate time when the optical flux starts to increase again.

First paragraph of Section 2: the authors write that they find “evidence for another decline until 140 days”. However, this statement does not seem supported by observations since the I-band light curve does not have data points between $t = 40$ days and $t = 140$ days. This point should be clarified.

A: The referee is right that the statement is not justified due to the lack of data points between $t = 50$ days and 140 days. During this period, it is possible that the optical emission would show further variability. We have reworded the text to better describe the evolution of optical light curve.

Second paragraph of Section 2: for clarity, the dips in the X-ray light curve should be described in this paragraph rather than in the third paragraph of Section 3. In addition, the meaning of the downward pointing arrows present in the lower panel of Fig. 1 should be explained in the text and in the caption of the figure.

A: Following the referee’s suggestions, we have moved third paragraph of Section 3 to Section 2, and merged it with the second half of paragraph 2 in Section 2. Now it is in a new paragraph, third paragraph in Section 2, where the unique properties in the X-ray light curve are better presented. The downward arrows are the 3σ upper limits on the flux for non-detections, which are explained in the text and in the caption of the Figure.

Section 3: the main goals of this section is determine the origin of the delayed X-ray emission compared to optical light and the dips in the X-ray light curve. To this end, the authors introduce several mechanisms where the optical emission is produced either by shocks or reprocessing and invoke the presence of a black

hole binary or a patchy absorber. However, the main conclusions of the analysis are unclear and I recommend to significantly reorganize the section to improve this point. The text should be modified to clearly present the different proposed scenarios and the ability of each of them to account for observations. The interpretations favored by the authors should be highlighted along with the main reasons for this choice. This revision is the most urgent to make about this section. In addition, more comments below aim at further improving specific aspects of the analysis.

A: Following the referee's suggestions, we have re-organized this section to better discuss possible mechanisms in interpreting our main findings, among which we converged towards the favored one as main conclusion. Detailed replies will be shown as below.

First paragraph of Section 3: the authors mention one possible origin for optical emission that results from shocks while the X-rays are produced by the accretion of the newly-formed disc. Within this picture, they propose that the delay of the X-rays is caused by the time required for the accretion disc to form. However, numerical simulations (e.g. Shiokawa et al. 2015; Bonnerot and Lu 2020) find that a peak of the shock-heating rate of $\sim 10^{44}$ erg/s requires collisions near the black hole that only take place once the assembly of the accretion disc has essentially completed. This seems to disfavor the late onset of the disc as the origin for the delayed X-rays, although the theoretical understanding of this process is still uncertain. Another perhaps more likely origin comes from the viscous timescale required for the disc to start accreting after it formed. Simulations suggest that it can reach a few fallback times t_0 because most of the gas is located at distances from the black hole much larger than the stellar pericenter. These different options should be discussed in the text when describing the scenario where optical emission is due to shocks.

A: We agree with the referee that such a scenario to interpret the delayed X-ray emission need to be discussed in more details. Indeed, the origin of the TDE optical emission is not yet clear. Piran et al. (2015, ApJ, 806, 164) proposed that the optical emission in TDEs is directly powered by the shock dissipation of orbital energy due to stream self-intersections near apocenter radius ($\sim a_{\text{min}}$). Although it roughly agrees with observations, the scenario is not verified as it is extrapolated from the simulations by Shiokawa et al. (2015, ApJ, 804, 85) which were carried out for a white dwarf disrupted by an intermediate mass black hole. The proposal for the delayed X-ray emission after the optical one is also based on the simulation result by Shiokawa et al. (2015), that accretion of the majority of the star's bound mass is delayed by a considerably longer time, i.e., by a factor of ~ 5 longer than the expectation from classical tidal disruptions.

On the other hand, as pointed out by the referee, with more realistic simulations of disk formation for a typical TDE like OGLE16aaa, Bonnerot & Lu (2020, MN, 495, 1374, Figure 16) demonstrated that the heating rate of the initial self-crossing shock is of $\sim 10^{44}$ erg/s at an intersection radius $R_{\text{int}} \sim 25 R_t$. Since this intersection radius is much smaller than the apocenter radius of $a_{\text{min}} \sim 85 R_t$ for their choice of parameters, the expected peak heating rate near the apocenter would be weaker than 10^{44} erg/s. Hence the stream collisions at apocenter, as proposed by Piran et al. (2015), alone may not be enough to account for the majority of optical emission in observations. The simulations by Bonnerot and Lu (2020) have also suggested that the gas leaving the initial intersection point will move inward, and then experience numerous secondary shocks with heating rate as high as of a few times 10^{44} erg/s, resulting in the rapid formation of a thick accretion disk at about $10 R_t$. A fraction of the gas from the disk can get soon accreted. In this picture, the proposed scenario for the delayed X-ray emission may not be favored. However, it should be noted that because of the complex dynamic process, how matter dissipates its orbital energy through stream collisions is still poorly understood, hence the heating rates lifted in simulations are uncertain. We have revised the text to better indicate the limit of the stream collisions and the scenario of delayed disk accretion in explaining the observations of OGLE16aaa.

The sentences on the size of X-ray emission have now moved to the end of fifth paragraph of Section 2 (Page 4). In addition, since the delayed disk accretion has been disfavored, the discussion in the second paragraph is not relevant anymore, hence deleted in the revision.

Third paragraph of Section 3: as explained above, most of this paragraph should be moved to Section 2 for clarity.

A: This paragraph has been moved to third paragraph in Section 2.

Fourth paragraph of Section 3: the term “fully consistent” seems to imply that the presence of a binary black hole is the only possible explanation for the dips seen in the X-rays. Because other possibilities such as that relying on a patchy absorber appear as likely to explain of this feature, these terms should be changed.

A: Following the referee’s suggestion, we have reworded it as “appears to be...”

Fifth paragraph of Section 3: this paragraph is dedicated to the scenario where the optical photons are produced by re-processing and should not rely on the presence of a black hole binary. As part of the general revision of this section recommended above, I suggest that the authors include the first sentence of this paragraph to the previous one that focuses on the scenario where a binary black hole is assumed.

A: We appreciate the referee's suggestion on the mix of content which makes us realize that we may not have made our point clear enough in the previous draft of the paper. We have re-organized the paragraph by moving the reprocessing scenario to the second paragraph, which continues the discussion on alternative scenario that is different from the stream-collisions as the main power source of optical emission. The paragraph is then followed by the discussion on the possibility of patchy obscuration to explain the unusual X-ray variability at later times, which we will show in detail in the following reply.

Fifth paragraph of Section 3: according to Fig. 4, the ratio of X-ray to optical luminosities is very close to one at peak. This value seems to represent a strong evidence for re-processing of the X-rays to be at the origin of the optical photons. If they were instead produced by shocks that are independent on the efficiency of accretion, there would be no reason for these two luminosities to be equal. This argument has for example been used by Van Velzen et al. (2020) when analyzing the X-ray flares seen from AT2019ehz, one of the seventeen TDE candidates detected by ZTF. It should be included in the revised version of the paper.

A: This paragraph has moved to second paragraph of Section 3 in the revision, and we have included the result of Figure 4 when discussing the reprocessing scenario. However, though the reprocessing (as proposed by Metzger & Stone 2016, MN, 461, 948) is able to explain the optical emission at early time as well as the feature of rapid X-ray brightening, it seems failed to reproduce the X-ray light curve at later times ($t > 200$ days), i.e., the epochs showing flux dips and flares. Similar X-ray flares have been observed in the TDE AT2019ehz (van Velzen et al. 2020), as pointed out by the referee, which can be explained by a patchy reprocessing layer that will lead to variable obscuration to X-ray source along the line of sight. However, the observed properties of OGLE16aaa appear different from AT2019ehz. The most distinct difference is that the UV/optical blackbody luminosity decays continuously in between the X-ray flares for AT2019ehz. In this case, the X-ray flux dips can be naturally explained by the obscuration where the reprocessing of accretion luminosity is important to continuously power the optical emission. For OGLE16aaa, as

Figure 1 shows, the optical emission decays quickly to a plateau phase after $t=146$ days, where the host emission dominates, suggesting that the reprocessing is less efficient. This is not consistent with the large optical depth required to obscure X-rays to explain the X-ray dip during the same period. Unfortunately, as we mentioned in the end of seventh paragraph (Page 8), because of the sparse UV observations, we lack of enough data points to confirm whether the UV emission is also largely reduced during this period. In addition, in the variable absorption scenario, the rapid brightening in the X-ray emission within only 10 days and lack of absorption feature in the X-ray spectrum between low and high flux state require extreme physical condition that the reprocessing layer near the gap only partially covers the X-ray source. Therefore, while the patchy obscuration remains a possible scenario, it may not be the most favorable one to explain the X-ray properties in OGLE16aaa.

Sixth paragraph of Section 3: the authors argue that the re-brightening of the optical light curve could be due to the presence of a black hole binary. However, other possibilities have been proposed in the context of ASASSN-15lh that only rely on a single black hole (Leloudas et al. 2016; Margutti et al. 2017). These other interpretations should be included along with potential reasons for favoring the scenario put forward by the authors.

A: We agree with the referee that the origin for the rebrightening in the light curve of ASASSN-15lh is debated. Leloudas et al. (2016) first proposed two emission mechanisms to explain the double-peak lightcurve of source. The rebrightening is due to the reprocessed accretion luminosity of TDE, which is delayed slightly to the emission at earlier times, while the initial peak is ascribed to the shock heating during stream collisions. Alternatively, Margutti et al. (2017) invoke a model in which a sudden change in the ejecta opacity responding to an underlying ionising source leads to the humped feature in lightcurve. However, Margutti et al. (2017) mainly focused on the UV emission and how to explain the optical re-brightening at longer wavelengths with their model is not well presented.

We argue that the optical rebrightening in OGLE16aaa is not due to the ionization “break out”, as it is more sensitive to the variability at the UV/X-ray bands. The reprocessing of accretion luminosity is likely but the rebrightening phase appears too short, requiring the accretion to be interrupted. This is not inconsistent with the SMBHB model because of the presence of many accretion islands in the early times, as shown in Figure 5. Unfortunately, we lack enough data points after the rebrightening to determine the further optical variability, hence better compare with the SMBHB model. Note that our case is different from the interpretation of the rebrightening in ASASSN-15lh by the TDE in an extreme mass ratio SMBHB, where the UV/optical emission is

supposed to originate directly from the accretion onto the secondary, instead of reprocessing (Coughlin et al. 2018, MN, 474, 3857). We have rephrased the paragraph to better indicate the favoring scenario for the optical re-brightening.

Seventh paragraph of Section 3: the authors propose that the dips in the X-rays is due to the passage of a dense gas cloud along the line of sight. They should explain the calculation made to obtain the value of $t_{\text{cross}} = 150$ days while the period of interruption can be as short as the exposure time of 1.7 ks used to obtain the X-ray upper limits with Swift.

A: This is indeed a concern and the description is opaque. In order to avoid confusion, we have now followed the calculation by van Velzen et al. (2020). Assuming that the distance of the intervening gas is the same as the radius of the photosphere for the UV/optical emission, which is $\sim 1-2 \times 10^{15}$ cm (0.4-0.8 light day) from the blackbody fittings, and a black hole mass of $10^{6.2} M_{\text{sun}}$ from Wyrzykowski et al. (2017), we obtained a t_{cross} of 50-140 days. This is comparable to the duration of X-ray non-detections (~ 140 days) between the optical peak and the first appearance of X-ray emission, as Figure 1 shows. Note that we lack enough data to determine accurately dip periods after the first X-ray peak, but under the assumption of orbital motion, the time intervals between subsequent X-ray flares are still consistent with each other. Therefore, we did not rely on the period of dip (duration of 1.7 ks) before the second X-ray flare to estimate the t_{cross} . We have rephrased the paragraph to clarify the calculation and comparison, and now presented in third paragraph of Section 3 in the revision.

Last paragraph of Section 3: as part of the revision of this section recommended above, this paragraph should be modified so that it clearly summarizes the main conclusions made by the authors regarding the interpretation of the observational data.

A: Following the referee's suggestion, we have modified the paragraph to more clearly summarize the main conclusions of our analysis.

Last paragraph of Section 3: the authors write that the origin of the delayed X-rays from a delayed disc assembly or accretion is disfavored based on the temporal evolution of the optical and X-ray signals but this claim is not justified in the paper. This statement should be motivated in the earlier parts of this section.

A: This sentence has been modified and moved to first paragraph in this

Section.

>>>> Minor comments (by order of appearance in the text)

Abstract: there appears to be words missing in the sentence containing “and promising prospect”. This should be fixed.

A: The sentence has been reworded. We have also checked through the text in the revision and tried our best to correct typos (highlighted in bold in the text).

Second paragraph of Section 1: there is a typo where “the first a few months” is written instead of “the first few months”. This should be corrected.

A: Corrected.

Second paragraph of Section 2: there is a typo where “reaching to an X-ray peak” is written instead of “reaching an X-ray peak”. This should be fixed.

A: Corrected.

Third paragraph of Section 2: there is a typo where “lack for emission” is written instead of “lack emission”. This should be fixed.

A: Corrected.

Fifth paragraph of Section 3: there are words missing in the sentence containing “X-rays is found about”. This should be corrected.

A: Corrected.

Note 1 of the supplementary information: a reference appears to be missing that has been replaced by a question mark. This should be fixed.

A: Corrected.

Caption of the supplementary Figure 4: the description of the top and bottom panels is inconsistent with the annotations in the plots referring to OGLE16aaa and J1201. This should be fixed.

A: Corrected.

Reviewer #3 (Remarks to the Author):

In their paper, 'An X-ray flare from the stellar tidal disruption by a candidate supermassive black hole binary,' the authors present an analysis of the optical/X-ray TDE candidate OGLE16aaa, and they investigate the possibility that the variability observed in the X-ray flux is due to the presence of a binary companion. They also consider the alternative scenario that the dips in the X-ray lightcurve are due to changes in the absorption along the line of sight.

Overall, I think the analysis carried out in this paper is interesting, and the interpretations seem broadly consistent with the observations. I have some comments/questions that I would like to see addressed before the paper is accepted for publication.

1) By-eye, the fit to the X-ray data in Figure 3 seems to over-represent the true flux at most wavebands. Is there a reason for this, i.e., is the fit being skewed by the one, apparent outlier point above the curve around ~ 0.8 keV? More specifically, it seems as though reducing the temperature by a modest factor would provide a much better fit for the data in the range between ~ 0.2 and 0.8 keV. The authors should verify their fit and/or provide an explanation for this apparent by-eye discrepancy.

A: We thank the referee to point this out. Our model fitting to the X-ray spectrum includes both the Galactic and intrinsic absorption, as mentioned in fourth paragraph of Section 2. The model shown in the Figure 3 is the *absorption-corrected* one, which we think better represents the intrinsic X-ray emission. Since the X-ray spectrum is soft for this observation, a single absorbed blackbody model can well describe the data. We have clarified it in the Figure caption.

2) The authors state that the I-band flux re-brightened after roughly a month. Investigating Figure 1, it appears as though there is tentative evidence for the UVW2 and UVM2 bands displaying a similar brightening. Is this statistically significant? If so, the authors should simply comment on this.

Figure A3: Left: Evolution of UVW2 flux as a function of time, fitted with a $t^{-5/3}$ (solid line) and exponential decay model (dashed line), respectively. The corresponding data/model ratio is shown in the lower panel. Right: The same as left, but for the optical I-band data.

A: As the referee suggested, the flux re-brightening is only tentative in the UVW2 and UVM2 bands. In fact, the trend depends strongly on how well the model fits to the lightcurve evolution. In addition to the canonical $t^{-5/3}$ decay law, we have also tried the fittings with an exponential model ($L \sim e^{-t/\tau}$), which has been found to better trace the luminosity evolution of some TDEs in the early phase (Holoien et al. 2014, MN, 445, 3253; 2016, MN, 455, 2918). The exponential model describes the data slightly better according to the chisq statistics (Fig A3, left). Note that due to a bug in our fitting code, the flux error of the host emission was not properly propagated when deriving the host-subtracted UV photometry in the previous draft of the paper, resulting in underestimate of flux errors. By taking into account this effect, the model fittings show that the UV re-brightening is in fact not significant, as shown in the lower panel of Fig. A3 (left), where the data/model ratio plot does not display tentative excess. Removing the data from the period of tentative re-brightening (from $t=41$ to $t=54$ days) yields only a slight difference in the reduced chisq, from $\text{chisq}/\text{dof}=38.6/21=1.83$ to $\text{chisq}/\text{dof}=30.8/18=1.71$, i.e., at a confidence level of about 90% for changing of 3 dof. For comparison, we performed similar analysis on the optical I-band data, and found that the re-brightening in the optical emission is indeed more obvious (Fig. A3, right). Therefore, we conclude that the tentative re-brightening in the UVW2 is not significant, which has now been commented at the end of first paragraph in Section 2. The analysis presented above is shown as a new paragraph in the Supplementary materials (Note 2 and Figure 2) in the revision.

3) Overall I agree with the authors that the X-ray luminosity is fairly erratic over the time that the system was observed, and in particular the extremely short rise time is uncharacteristic of a TDE. However, have the authors attempted to fit the

overall trend in the declining X-ray luminosity with a $t^{-5/3}$ power-law, similar to the optical data? If there were simply too much obscuring material at early times that prevented detection, the existence of such a power-law decline would hint at the validity of this argument (i.e., that it is the same underlying source powering the emission, and the X-rays were just obscured at early times).

Figure A4: The fit to X-ray lightcurve with a $t^{-5/3}$ power-law decay (black curve), assuming the same X-ray peak time as the optical one. The red curve is the best-fit to the UVW2 data, but scaled to the blackbody luminosity at $t=153$ days.

A: By assuming that the actual peak of X-ray emission is the same as that of optical one, i.e., accretion produces both the optical and X-ray emission soon after the TDE, we attempted to fit the $t^{-5/3}$ decline law to the X-ray data. In our fittings, we did not take into account the flux upper limits, as well as the data point for XMM1 that are too low than the expected evolution and may bias fittings. The result is shown in black line in Figure A4. While the overall X-ray evolution could be described by the $t^{-5/3}$ law, most observations at low flux state are clearly different from such a typical evolution, making the X-ray properties unique among known TDEs. Although the presence of obscuring material can explain the X-ray non-detections in the early stage, it is difficult to account for the flux dips and rapid rise to peak features observed at later times (at $t \sim 150$ and 350 days). Since the similar comparison has been shown in Figure 5, where our favored SMBHB scenario is presented, we prefer not to show in Figure 1 the fitting result with the $t^{-5/3}$ law for clarification. Interestingly, we found that the best-fit to the UVW2 data that is scaled to the UV/optical blackbody luminosity (red line in Figure A4) matches well with the above X-ray fitting with the $t^{-5/3}$ law. Such a result has been included in the lower panel of Figure 1 in the revision, which could be useful for a comparison of the X-ray luminosity with the UV/optical luminosity.

4) The authors state that the time to peak in the X-rays is given by the vertical, dashed, gray line in the bottom panel of Figure 1, which is constrained by the 4 very tightly spaced points around a time of ~ 140 days and in particular the 3 Swift X-ray observations. However, the difference in X-ray luminosity between the two points near the peak is extremely small, and then there appear to be no further observations between the last Swift point around 140 days and the non-detection near day 300. My question is therefore: how confident can the authors be that the true peak in the X-rays was not significantly later than the claimed peak, given the lack of observations? It would help if the authors could constrain the average decline per point 3 above, which would potentially constrain the location of the true peak if one assumes a $t^{-5/3}$ decline.

A: We agree with the referee's point that the true peak in the X-rays cannot be tightly constrained with current data, which are sparsely sampled. Strictly speaking, our claim was referred to the time of maximum luminosity we could observe. However, because the maximum luminosity at 0.3-2 keV is about 7×10^{43} erg/s, corresponding to an integrated blackbody luminosity of $\sim 4 \times 10^{44}$ erg/s, comparable to the Eddington luminosity for a $\sim 2 \times 10^6$ solar mass black hole of OGLE16aaa (Wrzykowski et al. 2017). So if the true peak time were later than we claimed, the difference should be small. Indeed, as suggested by referee, by fitting a $t^{-5/3}$ decline to the X-ray evolution (see our reply to question #3 above), we do not find significant derivation of claimed peak luminosity from the expected. In the revision, we have commented the uncertainty in our claimed X-ray peak in the caption of Figure 1.

5) In Figure 2, the authors should show at least one of the results of fitting the additional power-law/blackbody models to the data in addition to the ratio of the data to the model (i.e., similar to the top panel of this figure; this could also conceivably be achieved by simply adding another line that includes the power-law/blackbody component to the top panel of this figure).

A: Following the referee's suggestion, we have added a new panel (a) in Figure 2 to show the spectral fitting result with an additional blackbody component for the XMM2 data. The following panels have been re-arranged accordingly.

6) The authors propose that the declines in X-ray flux are consistent with the predictions from the scenario of the disruption of a star by a SMBH binary. However, the theoretical arguments behind this prediction are that the X-ray luminosity scales with the fallback rate, which is temporarily interrupted as the secondary perturbs the orbits of the infalling debris. It is therefore not obvious to me as to why *only* the X-ray emission would be stunted/interrupted in this model; why wouldn't the UV/optical also show some signs of interruption if the

flow and the resulting accretion luminosity are driven by the returning debris? Some explanation along these lines should be provided.

A: We agree with the referee that besides the X-ray emission, UV/optical emission would be also interrupted if originated from accretion luminosity. However, as shown in Figure 3, extrapolating the best-fitting blackbody model that describes the X-ray spectrum well to the UV/optical bands leads to a severe underestimate of observed emission by at least 2 orders of magnitude. This is likely due to the low black hole mass of the system hence shift of the peak of accretion luminosity to the soft X-rays. The UV/optical emission from accretion disk appears too low compared with the observed luminosities. Alternative origins for the observed UV/optical emission could be the shocking heating by stream self-collisions near apocentre, or the reprocessing of accretion luminosity by a radiation ejecta. The latter is our favorable scenario to explain optical re-brightening in OGLE16aaa. As we suggested in Section 3 (third paragraph, Page 7), in the reprocessing picture, the relatively short-lived optical re-brightening may link to the discrete accretion (while the X-ray radiation is obscured), which is not at odds with the prediction of the SMBHB model. For $t > 140$ days, however, the I-band flux might have fallen below the host level, and the data in UV are too sparse, so it is not so certain to say whether they are synchronous with X-rays or not.

7) In the binary disruption scenario, in addition to the accretion rate onto the primary being reduced, there can also be an increase in the accretion rate onto the secondary as it strongly interacts with the stream of debris produced by the disruption; based on the models of Ricarte et al. (2016) (<https://ui.adsabs.harvard.edu/abs/2016MNRAS.458.1712R/abstract>) and Coughlin et al. (2017) (<https://ui.adsabs.harvard.edu/abs/2017MNRAS.465.3840C/abstract>), this becomes increasingly likely as the mass ratio of the binary nears unity, and certainly seems like it should be possible for the best-fit mass ratio of $q=0.25$ adopted by the authors. Is this additional reservoir of accretion energy accounted for by the model employed by the authors? The authors should at least mention the possibility that a contribution to the X-ray flux (and presumably other wavelengths) could be provided by the accreting secondary.

A: In our model, the separation of two SMBHs is relatively large compared to the semi-major axis of the most bound debris. In this configuration, most of materials will be accreted by the primary BH. Only a minor fraction will be accreted by the secondary. Actually, Ricarte et al. (2016) has already pointed out that only if the binary SMBH is "in extremely tight and equal-mass

configurations", the contribution of the secondary would be significant, which is however, not the case in our model. In addition, based on Fig. 12, Fig. 13 and Fig. 18 of Coughlin et al. (2017), which has more close configurations as we adopted, the averaged contribution of accretion rate from the secondary BH, in most of cases, is at least an order of magnitude lower than the primary one. So we think the contribution of accretion luminosity from the secondary BH can be neglected in our model. This point has been mentioned in the revision (first paragraph of Page 7) when discussing the possible origins for the hard excess in the X-ray spectrum.

8) It would be useful if the authors included a diagram that indicates the qualitative features of their favored scenario, i.e., an envelope that reprocesses/obscures the X-rays to some time, and a binary companion that induces variability in the fallback rate onto the primary.

A: Following the referee's suggestion, we present in Figure 5 a schematic illustration of the tidal disruption by SMBHB for the origin of the X-ray variability, along with the obscuration in the early phase. The features of evolution at different stages are described in the caption.

9) In their conclusions, the authors should note the gravitational-wave inspiral time of the SMBH binary used in their model for the fallback.

A: Given the mpc separation of the SMBHB in our model, the effect of gravitational-wave emission on the orbital parameters and shrinkage is negligible. Based on the best-fit SMBHB model parameters in Table 3, we estimated the gravitational-wave inspiral time $t_{gw} \sim 1.9e7$ years (Peters & Mathews 1963, PhRv, 131, 435), which is much longer than the timescale of a few years for the TDE of our interest. We have added a new sentence to explicitly note that in its current state of evolution, it would be challenging to detect the GWs from the SMBHB system.

REVIEWER COMMENTS

Reviewer #1 (Remarks to the Author):

I would like to thank the Authors for a clear and thorough response. Their corrections to the Manuscript are satisfying to me.

Reviewer #2 (Remarks to the Author):

The authors have satisfyingly addressed the suggestions made in the first referee report. I have still the important comment below concerning a part of the text that should be modified. When this is done, the paper can be accepted for publication.

First paragraph of Section 3: The authors should rewrite the passage “However, recent simulations ... onset of accretion” since it contains several incorrect statements. I reformulate below the arguments disfavoring a delayed onset of disc formation as the origin of the early lack of X-rays. Although this has been proposed by Piran et al. (2015) from analytical arguments, it appears unlikely that shocks near apocenter lead to high optical luminosity. This is argued in the more recent work by Lu and Bonnerot (2020) based on the fact that large optical depths surrounding the self-crossing shock prevent most of the dissipated energy from promptly emerging from the gas. Instead, numerical simulations (Shiokawa et al. 2015; Bonnerot and Lu 2020) suggest that bright optical emission is only expected once the debris has reached a significant level of circularization due to strong collisions taking place near pericenter. Because such powerful optical emission is detected at early times from OGLE16aaa, it is likely that the gas has already significantly circularized, which disfavors the interpretation that the absence of X-rays is due to a slow disc formation.

Reviewer #3 (Remarks to the Author):

I thank the authors for responding to my queries and suggestions. I now find the paper suitable for publication.

Dear Referees,

We thank you again for taking the time to carefully review the revised version of the manuscript. As in our first round of reply, we describe below our replies (in blue) to take into account the comment raised by the Referee #2 that is helpful to improve the manuscript. Relevant changes in the main text are shown in bold font (based on the last revised version).

Reviewer #2 (Remarks to the Author):

First paragraph of Section 3: The authors should rewrite the passage “However, recent simulations ... onset of accretion” since it contains several incorrect statements. I reformulate below the arguments disfavoring a delayed onset of disc formation as the origin of the early lack of X-rays. Although this has been proposed by Piran et al. (2015) from analytical arguments, it appears unlikely that shocks near apocenter lead to high optical luminosity. This is argued in the more recent work by Lu and Bonnerot (2020) based on the fact that large optical depths surrounding the self-crossing shock prevent most of the dissipated energy from promptly emerging from the gas. Instead, numerical simulations (Shiokawa et al. 2015; Bonnerot and Lu 2020) suggest that bright optical emission is only expected once the debris has reached a significant level of circularization due to strong collisions taking place near pericenter. Because such powerful optical emission is detected at early times from OGLE16aaa, it is likely that the gas has already significantly circularized, which disfavors the interpretation that the absence of X-rays is due to a slow disc formation.

A: We agree with the referee’s argument that shock heating near apocenter appears too weak to power the observed optical emission in most TDEs. To account for observations requires the shock heating rate at the order of a few times 10^{44} erg/s, which can only take place close to pericenter where the debris self-collisions and energy dissipation are strong, as implied by the recent simulations (Bonnerot & Lu 2020). In this case, the circularization process will be efficient as well, leading to rapid formation of disk. Hence the scenario of late time X-ray brightening due to delayed disk formation seems disfavored. Although such a point has been outlined in our first round of reply, it may not be well explained in the main text. Following the referee’s suggestion, we have now reworded the relevant sentences in the first paragraph of Section 3 (Page 5) to clarify this part of discussion.

REVIEWERS' COMMENTS

Reviewer #2 (Remarks to the Author):

I think that the paper is in good enough shape now for it to be accepted for publication.